# Reinforcement Learning with Verifiable Rewards Implicitly Incentivizes Correct Reasoning in Base LLMs

**Xumeng Wen**[*1], **Zihan Liu**[*†2], **Shun Zheng**[*‡1], **Shengyu Ye**[†1], **Zhirong Wu**[1], **Yang Wang**[1],
**Zhijian Xu**[†3], **Xiao Liang**[†4], **Junjie Li**[1], **Ziming Miao**[1], **Jiang Bian**[1], **Mao Yang**[1]
[1]Microsoft Research Asia [2]Peking University
[3]The Chinese University of Hong Kong [4]University of California, Los Angeles

## Abstract

Recent advancements in long chain-of-thought (CoT) reasoning, particularly through the Group Relative Policy Optimization algorithm used by DeepSeek-R1, have led to significant interest in the potential of Reinforcement Learning with Verifiable Rewards (RLVR) for Large Language Models (LLMs). While RLVR promises to improve reasoning by allowing models to learn from free exploration, there remains debate over whether it truly enhances reasoning abilities or simply boosts sampling efficiency. This paper demonstrates the profound impact that RLVR has on the reasoning capabilities of LLMs. We revisit Pass@K experiments and show that RLVR can extend the reasoning boundary for both mathematical and coding tasks. This is supported by our introduction of a novel evaluation metric, CoT-Pass@K, which captures reasoning success by accounting for both the final answer and intermediate reasoning steps. Furthermore, we present a theoretical framework explaining RLVR's incentive mechanism, demonstrating how it can encourage correct reasoning even when rewards are based solely on answer correctness. Our analysis of RLVR's training dynamics reveals that it incentivizes correct reasoning early in the process, with substantial improvements in reasoning quality confirmed through extensive evaluations. These findings provide strong evidence of RLVR's potential to enhance LLM reasoning, offering valuable insights into its mechanisms and performance improvements.

## 1 Introduction

The successful replication of long chain-of-thought (CoT) reasoning, similar to that in OpenAI's o1 (OpenAI, 2024), by DeepSeek-R1 (Guo et al., 2025) using the Group Relative Policy Optimization (GRPO) algorithm (Shao et al., 2024), has sparked a surge of interest within the open research community. This interest is focused on understanding, reproducing, and extending DeepSeek's approach, as evidenced by a multitude of recent studies (Liu et al., 2025b; Hu et al., 2025; Zeng et al., 2025; Yu et al., 2025; He et al., 2025; Wen et al., 2025; Chen et al., 2025b). Fundamentally, this emerging paradigm is a form of Reinforcement Learning with Verifiable Rewards (RLVR) (Lambert et al., 2024; Guo et al., 2025; Yue et al., 2025), where a Large Language Model (LLM) acts as a policy, generating a CoT as a sequence of actions and receiving feedback on answer correctness from deterministic verifiers. This paradigm holds the promise of endowing LLMs with the ability to learn from experience through free exploration, potentially leading to unlimited intelligence (OpenAI, 2024; Guo et al., 2025; Silver & Sutton, 2025).

However, emerging concerns question the true effectiveness of RLVR. These concerns are motivated by the observation that some post-RLVR models improve the Pass@1 metric but fail to enhance the Pass@K metric compared to the base (pre-RLVR) model. This phenomenon was first noted by Shao et al. (2024) during the development of GRPO. Subsequently, a systematic study by Yue et al. (2025)

---

[*]These authors contributed equally: Xumeng Wen, Zihan Liu, Shun Zheng.

[†]Work done during the internship at Microsoft Research Asia.

[‡]Correspondence to shun.zheng@microsoft.com.

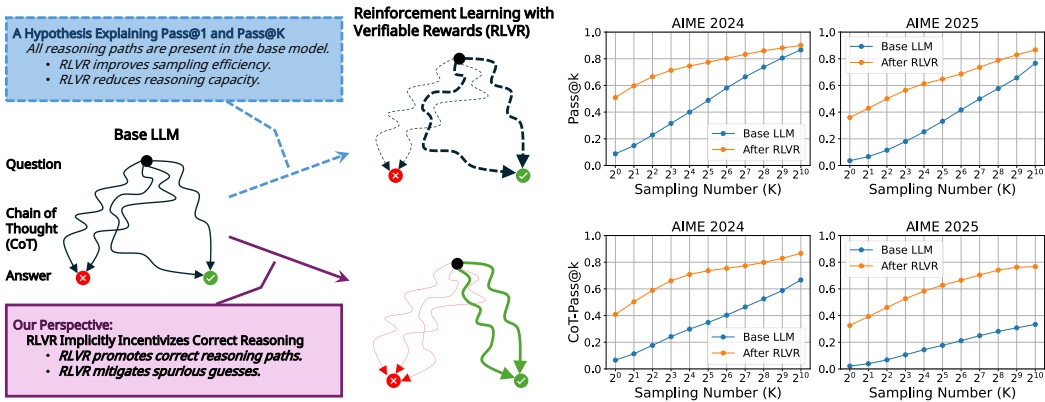

Figure 1: An illustration of our perspective: **RLVR implicitly incentivizes correct reasoning in base LLMs**. We visualize how different explanation frameworks lead to varying reasoning paths being activated, with our perspective shown in the lower left and a recent popular hypothesis explaining Pass@K observations (Yue et al., 2025) summarized in the upper left. In this diagram, the line width represents the sampling probability of a reasoning path, while the color distinguishes correct paths (green) from incorrect ones (red). If all reasoning paths after applying RLVR are already present in the base model, the reasoning model merely adjusts the sampling probabilities of these existing paths (visualized in dashed lines). This hypothesis effectively accounts for the key observation shown in the upper-right part, where, for a moderately large $K$, a base LLM can catch up to the reasoning model after RLVR using the Pass@K metric. In this study, we unveil **the extended reasoning capability boundary** in math tasks using a refined metric, **CoT-Pass@K**, which emphasizes both the correctness of answers and the validity of reasoning CoTs. Verification results supporting this figure are publicly available at `https://huggingface.co/datasets/XumengWen/AIME24-25_CoT_Verification`.

on more open-weight RLVR models discovered that the Pass@K metric of the base model increases at a much faster rate than its RLVR-tuned counterpart. Consequently, for a moderately large K, the base model eventually matches and surpasses the reasoning model. This led to their adventurous hypothesis: all correct reasoning paths are already present in the base model, and RLVR merely improves sampling efficiency at the cost of reducing overall reasoning capacity.

While this hypothesis has gained significant support (Zhu et al., 2025; Zhang et al., 2025; Wang et al., 2025a; Chen et al., 2025a), conflicting observations have also been reported. For instance, Liu et al. (2025a) detected the emergence of new reasoning patterns after RLVR, while they also acknowledged a loss in reasoning capacity as measured by Pass@K. Chen et al. (2025b) reported persistent improvements in Pass@K for competitive coding tasks but did not show improved Pass@K for math tasks. Shojaee et al. (2025) observed similar Pass@K observations on math datasets but found different patterns on puzzles with high complexity. To the best of our knowledge, no systematic explanation exists to reconcile these contradictory findings, leaving a critical question unanswered: "*should we accept the hypothesis as a fundamental limitation of RLVR or should we trust new empirical findings that challenge the hypothesis?*"

In this work, we address this debate systematically and demonstrate that RLVR can fundamentally enhance the reasoning abilities of LLMs. First, we revisit Pass@K experiments and unveil the existence of extended reasoning capability boundaries after RLVR for both math and code tasks. In addition to reproducing the extended reasoning boundary in competitive coding, as reported by Chen et al. (2025b), we find that the Pass@K performance of base LLMs on math reasoning can be unreliable, as base LLMs are capable of producing incorrect CoTs yet coincidentally arriving at the ground truth, especially for hard mathematical questions where answers are simple and can be easily guessed after multiple attempts. To address this, we introduce a new metric, CoT-Pass@K, which evaluates success only when both the final answer and the intermediate reasoning CoT are correct. In practice, we verify the correctness of mathematical CoTs by instructing DeepSeek-R1-0528-Qwen3-8B (DeepSeek, 2025) and confirm their reliability. Using this new metric, we successfully identify

the extended reasoning boundary of a post-RLVR model for math tasks. Figure 1 summarizes our key perspectives.

Moreover, we develop a theoretical framework to explain why RLVR works, even when base LLMs may guess the ground truth and only answer correctness is provided as a reward, and how RLVR incentivizes correct reasoning. Our central insight is that once LLMs have been pre-trained to establish strong knowledge and logic priors that distinguish correct from incorrect CoTs, the GRPO gradient will increase the probability of generating more correct CoTs.

Additionally, we investigate the training dynamics of RLVR to understand when this improved reasoning emerges. By reproducing GRPO-style training using the open-source DAPO recipe (Yu et al., 2025) and performing extensive verifications, we find that RLVR begins to incentivize correct reasoning from the early stages of training, and this capability generalizes well to unseen test questions. The results of our training analysis align with our theorem, which highlights the implicit incentivization of correct reasoning CoTs.

Finally, we evaluate the quality of generated CoTs from a learning perspective: if supervised learning on some CoT data results in better generalization performance on test sets, we regard them as high quality. This allows us to evaluate the quality of CoTs generated by model checkpoints at different RLVR stages. Our results show that after RLVR, the quality of reasoning CoTs has been fundamentally improved.

In summary, our contributions include:

- A systematic evaluation revealing the extended reasoning capability boundary after RLVR for both code and math tasks.

- A theoretical understanding of why RLVR works with only answer correctness as a reward and how RLVR incentivizes correct reasoning.

- An analysis of RLVR's training dynamics, delving deeper into optimization effects, generalization behaviors, and current limitations.

- Confirmation of the quality improvements in reasoning CoTs from a learning perspective, replicating the generalization abilities of post-RLVR models trained with enormous costs simply via supervised fine-tuning.

## 2 RELATED WORK

**RLVR** Since the release of DeepSeek-R1 (Guo et al., 2025), there has been a surge of research interest in the RLVR paradigm (Luo et al., 2025b; Liu et al., 2025b; Hu et al., 2025; Cui et al., 2025; Xie et al., 2025; Zeng et al., 2025; Yu et al., 2025; Luo et al., 2025a; Chen et al., 2025a; He et al., 2025; Wen et al., 2025; Cao et al., 2025; Liu et al., 2025a; Chen et al., 2025b). Due to the high computational cost of RLVR, most studies have focused on small- to medium-sized models (up to 32B parameters). These studies span a wide range of aspects, including training data curation, objective design, hyperparameter tuning, base model selection, and various insightful observations. However, only a few studies have addressed the theoretical foundations of RLVR. In this work, we argue that RLVR for LLMs should be understood from a different perspective—one that emphasizes the correctness of reasoning paths. We hope our empirical findings and theoretical perspective could inspire the community to develop more efficient and effective RLVR approaches, unlocking its broader potential across diverse applications.

**Debates on Whether RLVR Really Incentivizes** Since Yue et al. (2025) raised the insightful question of whether RLVR truly incentivizes improvements beyond the base LLMs, and conducted extensive empirical experiments to demonstrate the wide applicability of their key hypothesis—that RLVR does not improve $Pass@K$ for the base LLM because all reasoning paths are already present in the base model—there have been varying perspectives on this hypothesis. Some studies agree with this viewpoint (Wang et al., 2025b; Zhu et al., 2025; Zhang et al., 2025; Wang et al., 2025a; Chen et al., 2025a), while others report contradictory findings (Liu et al., 2025a; Chen et al., 2025b; Shojaee et al., 2025), as discussed in the introduction. There is currently no fundamental understanding to resolve these debates. Liu et al. (2025a) speculated that previous RLVR experiments

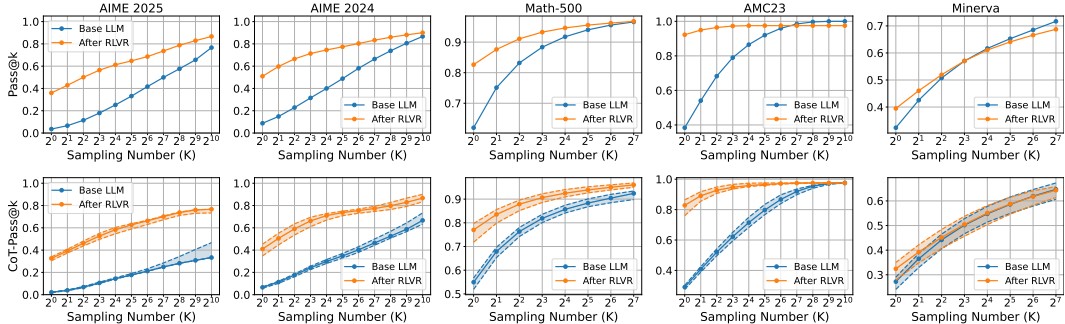

Figure 2: Comparisons of Pass@K (the top row) and CoT-Pass@K (the bottom row) on five math benchmarks (different columns) to show how RLVR could improve base LLMs. Here the base LLM is Qwen2.5-32B, and the post-RLVR model is DAPO-Qwen-32B. For CoT-Pass@K, we perform multiple verifications for each CoT using DeepSeek-R1-0528-Qwen3-8B, and display the results determined by *any-correct*, *all-correct*, and *majority-correct* strategies, which constitute the shaded area in lower subplots.

may have been conducted within a single domain (e.g., math) and were optimized for limited gradient steps before true exploration could occur. Shojaee et al. (2025) suggested that the complexity of puzzles might be the key factor. Chen et al. (2025b) presented statistically significant empirical results to justify that their model indeed improves $Pass@K$, particularly highlighting a persistent gap on the LiveCodeBench v6 (Jain et al., 2025). In this work, we extend the Pass@K experiments of Chen et al. (2025b) to more LiveCodeBench versions and include another similar open-source study (He et al., 2025) starting RLVR from distilled LLMs as a comparison. All these results clearly disclose the extended reasoning boundary of distilled LLMs on competitive coding after RLVR.

**The Importance of Correct CoTs**   Recent studies have also highlighted the importance of verifying the correctness of CoTs (Arcuschin et al., 2025; McGinness & Baumgartner, 2025; Shojaee et al., 2025). However, their approaches focus on defining synthetic reasoning tasks where the correctness of reasoning CoTs can be verified easily. While this is an interesting and effective approach for fully examining reasoning correctness, it is difficult to apply to unstructured reasoning scenarios, such as in math and code. In this work, we argue that the LLM-as-a-CoT-Judge paradigm could play a crucial role in more general reasoning tasks, and emphasize the pressing need for the design of evaluation benchmarks to assess the reliability of emerging LLM verifiers.

## 3   EXTENDED REASONING CAPABILITY BOUNDARY AFTER RLVR

In this section, we present concrete benchmark evaluations that demonstrate how RLVR can fundamentally enhance the reasoning abilities of LLMs. This enhancement goes beyond mere improvements in sampling efficiency; it also expands the reasoning capability boundary. However, to effectively observe this enhancement, it is crucial to adopt an appropriate RLVR training recipe, select challenging benchmarks that are free from data contamination, and utilize reliable evaluation metrics. Without these measures, one might only observe improvements in sampling efficiency, with no actual change in reasoning capacity. Below, we discuss two representative cases from both the math and code domains, showcasing genuinely extended reasoning boundaries.

### 3.1   MATH REASONING

We begin by revisiting the Pass@K experiments conducted on the open-source model, DAPO-Qwen-32B (Yu et al., 2025), which successfully reproduced R1-Zero (Guo et al., 2025) using the base LLM, Qwen2.5-32B (Qwen, 2024), and a curated set of $17k$ mathematical problems. A key contribution of our work is the introduction of a novel evaluation metric, CoT-Pass@K, which emphasizes the importance of evaluating the correctness of detailed reasoning steps for mathematical questions, rather than relying solely on answer correctness.

Figure 3: Comparisons of Pass@K across six LiveCodeBench versions to show how much RLVR could enhance distilled LLMs. Here the distilled LLM is DeepSeek-R1-Distill-Qwen-7B, and the post-RLVR model is AceReason-Nemotron-7B.

However, precisely measuring the correctness of CoTs at scale is inherently challenging due to the unstructured, lengthy, and complex nature of math CoTs. Fortunately, we can leverage specialized open-source LLMs, such as DeepSeek-R1-0528-Qwen3-8B (DeepSeek, 2025), as a powerful yet lightweight verifier, employing an LLM-as-a-CoT-Judge paradigm. In this study, we use this verifier multiple times for each reasoning CoT generated by DAPO-Qwen-32B and Qwen2.5-32B, employing three distinct strategies to assess CoT correctness: *any-correct* (at least one verification returns correct), *all-correct* (all verifications must return correct), and *majority-correct* (the majority vote determines the outcome). To further ensure the reliability of these verifications, we manually inspect cases where the Pass@K metric yields a small positive value, but the CoT-Pass@K metric returns zero. The details of our LLM-as-a-CoT-Judge approach can be found in Appendix A.2.

Figure 2 presents a comprehensive comparison using both Pass@K and CoT-Pass@K metrics across prominent math-reasoning benchmarks. As shown in the top row, the Pass@K results align with the observations in (Yue et al., 2025): the performance of the base LLM quickly catches up with and even surpasses the post-RLVR model as $K$ increases. However, in stark contrast, the CoT-Pass@K results on AIME 2024 and AIME 2025 reveal a consistent and significant performance gap between the models across all values of $K$ (up to 1024). This gap is particularly pronounced on AIME 2025, possibly due to its complete absence of unintentional data contamination, as it was released after the base model's training cutoff. We perform manual inspections to ensure the distinct gaps observed using the CoT-Pass@K metric are reliable (see examples in Appendix A.7.1 and A.7.2). Our LLM verifier effectively identifies critical errors, which we agree should be rejected. These results demonstrate the extended reasoning boundary of DAPO-Qwen-32B over Qwen2.5-32B.

Additionally, we observe that on other benchmarks such as MATH-500 and AMC23, the effects of RLVR seem less pronounced, as the base LLM is already capable of solving these problems correctly with sufficient attempts. This may be due to 1) the problems being simple enough for the base LLM to solve using its existing knowledge, or 2) the problems being part of its pre-training data, allowing the base LLM to recall the correct solution after multiple trials. Distinguishing between these possibilities is challenging without knowing the exact training data used for Qwen2.5-32B. Furthermore, on the Minerva benchmark, the post-RLVR model shows no improvement, likely due to a train-test domain mismatch. Minerva contains numerous physics problems and free-form answers, while the DAPO training data was restricted to math problems with integer answers. These results do not undermine the effectiveness of RLVR; rather, they highlight the importance of selecting appropriate benchmarks for evaluating RLVR progress.

## 3.2 CODE REASONING

Unlike RLVR for mathematical problems, where the correctness of extracted answer tokens is used as a proxy for passing, code reasoning relies on the actual execution of generated code snippets to verify correctness, significantly reducing the likelihood of guessing. Therefore, Pass@K serves as a reliable evaluation metric for code reasoning tasks.

In this section, we reproduce the Pass@K experiments across different versions of Live-CodeBench (Jain et al., 2025) to compare the performance of AceReason-Nemotron-7B (Chen et al., 2025b) with its pre-RLVR counterpart, DeepSeek-R1-Distill-Qwen-7B (Guo et al., 2025). As shown in Figure 3, we observe that AceReason-Nemotron-7B exhibits clear Pass@K improvements over DeepSeek-R1-Distill-Qwen-7B on most benchmark versions, even though the latter is a distillation model already demonstrating remarkable reasoning capabilities. These results suggest that even

for distillation models, a high-quality RLVR training recipe can significantly extend the reasoning capability boundary, particularly for competitive coding tasks.

To further confirm the success of RLVR in extending reasoning boundaries for coding tasks, we evaluate another post-RLVR model, Skywork-OR1 (He et al., 2025), which has a fully reproducible training recipe publicly available. Detailed results on LiveCodeBench-v6 can be found in Appendix A.3, where we observe a consistent Pass@K gap between Skywork-OR1 and DeepSeek-R1-Distill-Qwen-7B. Specifically, we find that only medium and hard problems in LiveCodeBench-v6 contribute to the differentiation between these two models for large $K$ values, underscoring the importance of selecting challenging benchmarks.

# 4 A THEORETICAL UNDERSTANDING OF RLVR FOR LLMS

In addition to empirical evidences, we provide a theoretical understanding of how RLVR, as implemented in the GRPO algorithm (Shao et al., 2024), fundamentally incentivizes correct reasoning for pre-trained language models. We note a key distinction between RLVR for LLMs and traditional RL for randomly initialized models. Pre-trained LLMs, owing to their powerful likelihood estimation capabilities obtained during pre-training, can generate various CoTs and then produce possible answers. Some of them could coincidentally arrive at the ground truth, especially when the ground truth is in a simple format and can be easily guessed. In contrast, traditional RL simply optimizes for action trajectories that yield high rewards, without necessarily verifying the intrinsic correctness of each action along the path. For instance, in the Go game (Silver et al., 2017), every action is valid once the simulation environment is setup correctly. In the following, we start our theoretical analysis from a formal problem setup distinguishing CoT and answer tokens in LLM responses.

**Problem Setup**   Given a question prompt $q$, we sample $G$ responses $\mathbf{Y} = \{y_1, y_2, \ldots, y_G\}$ from policy $\pi_\theta$, where $\pi_\theta$ is a LLM model parameterized by $\theta$. Let $c_i$ be the CoT in response $y_i$, and $a_i$ the final answer. We define the following correctness indicators:

$$\mathcal{I}_{\text{CoT}}(c_i) = \begin{cases} 1 & \text{if } c_i \text{ is correct} \\ 0 & \text{otherwise} \end{cases}, \quad \mathcal{I}_{\text{Ans}}(a_i) = \begin{cases} 1 & \text{if } a_i \text{ is correct} \\ 0 & \text{otherwise} \end{cases}. \tag{1}$$

In this study, we define the CoT correctness $\mathcal{I}_{\text{CoT}}(c_i)$ as the intermediate tokens of a response $(c_i)$ expressing necessary and accurate logics that lead to the ground truth. We use $p_c^\theta = P_{\pi_\theta}(\mathcal{I}_{\text{CoT}}(c) = 1)$ to denote the probability of generating a correct CoT. In practice, it is rather challenging to verify the CoT correctness because it is inherently unstructured, knowledge-intensive, and full of details. In contrast, the answer correctness $\mathcal{I}_{\text{Ans}}(a_i)$ is assumed to be verified programmatically. So we have a verifiable reward $R(y_i)$ that is binary and determined solely by answer correctness: $R(y_i) = \mathcal{I}_{\text{Ans}}(a_i)$. We calculate the standard GRPO advantage $\hat{A}(y_i)$ as:

$$\hat{A}(y_i) = \frac{R(y_i) - \mu_{\mathbf{Y}}}{\sigma_{\mathbf{Y}}}, \quad \mu_{\mathbf{Y}} = \frac{1}{G}\sum_{j=1}^{G} R(y_j), \quad \sigma_{\mathbf{Y}} = \sqrt{\frac{1}{G}\sum_{j=1}^{G}(R(y_j) - \mu_{\mathbf{Y}})^2}. \tag{2}$$

Without loss of generality, we consider a policy gradient (Sutton et al., 1999) update:

$$\nabla_\theta J(\theta) \approx \frac{1}{G}\sum_{i=1}^{G} \hat{A}(y_i)\nabla_\theta \log \pi_\theta(y_i \mid q). \tag{3}$$

**Assumptions**   Given the problem setup decoupling CoT and answer correctness, we introduce a critical *Logic Prior* assumption: compared with incorrect CoTs, correct CoTs have higher probabilities to induce correct answers. Thus we have

$$P(\mathcal{I}_{\text{Ans}}(a_i) = 1 \mid \mathcal{I}_{\text{CoT}}(c_i) = 1) = \alpha > P(\mathcal{I}_{\text{Ans}}(a_i) = 1 \mid \mathcal{I}_{\text{CoT}}(c_i) = 0) = \beta. \tag{4}$$

This assumption is based on the belief that pre-trained LLMs have established strong knowledge and logic priors. Besides, we also assume a learnable group ($\sigma_{\mathbf{Y}} > 0$) and a sufficiently large sampling number $G$ to ensure stable gradient updates. Then, we establish the following theorem.

**Theorem 1 (GRPO Implicitly Incentivizes Correct Reasoning)** *For any prompt $q$ satisfying our assumptions, the expected GRPO advantage $\mathbb{E}[\hat{A}(y_i)]$ satisfies:*

$$\mathbb{E}\left[\hat{A}(y_i) \mid \mathcal{I}_{CoT}(c_i) = 1\right] > 0, \quad \mathbb{E}\left[\hat{A}(y_i) \mid \mathcal{I}_{CoT}(c_i) = 0\right] < 0, \tag{5}$$

*where $\hat{A}(y_i)$ is defined in equation 2. The GRPO policy gradient, as defined in equation 3, increase the probability of generating correct CoTs ($p_c^\theta$) in the next round, so $p_c^\theta$ increases monotonically.*

Below we briefly illustrate our key perspectives on why GRPO works and when it may fail. Moreover, we include a complete proof for Theorem 1 and more discussions in Appendix A.4.

**Discussions on the effectiveness of GRPO**  Theorem 1 indicates that even though a base LLM may guess the ground truth with imperfect CoTs at the beginning (low initial $p_c^\theta$), GRPO could still work as long as the knowledge and logic priors have been established. The driving factor is the gap $\alpha - \beta > 0$, which amplifies the advantage difference between correct and incorrect CoTs. As training progresses and $\alpha$ increases (due to more sound reasoning across various question prompts) while $\beta$ decreases (reducing spurious correlations, model biases, incorrect knowledge or calculation, etc.), causing the gap to widen and further accelerating coherent reasoning. As $p_c \to 1$, $(\alpha - \beta)$ may approach 1 in a faster pace because generating a few answer tokens is typically much easier than producing long correct CoTs, then $\mathbb{E}[\hat{A}(y_i) \mid \text{correct CoT}] \to 0$, ensuring convergence.

**Discussions on failure modes in GRPO**  We note that the *Logic Prior* assumption may not always hold, potentially leading to the reinforcement of incorrect CoTs, since base LLMs may retain inherent biases and possibly fatal knowledge errors from pre-training. These harmful information might exist in some CoTs that finally yield the correct answer. In such cases, improper model biases could be unintentionally reinforced. We suspect that these unexpectedly reinforced CoTs are the root cause of the challenges faced by the R1-Zero approach (Guo et al., 2025), including poor readability and multi-lingual behaviors.

## 5  TRAINING DYNAMICS OF RLVR

To further demystify RLVR, we reproduce and analyze the training recipe of DAPO (Yu et al., 2025), which has been demonstrated to present extended reasoning capability boundaries in Section 3. Our experiments show that its training dynamics align pretty well with Theorem 1.

**Key Indicators**  We first introduce key indicators that we have recorded during the reproduction. For each prompt $q$ sampled with $G$ responses, we define the number of answer passes and the number of both CoT and answer passes as $C = \sum_{i=1}^{G} \mathcal{I}_{\text{Ans}}(a_i)$ and $D = \sum_{i=1}^{G} \mathcal{I}_{\text{CoT}}(c_i) \cdot \mathcal{I}_{\text{Ans}}(a_i)$, respectively. We follow Chen et al. (2021)'s approach to calculate the Pass@K metric. Accordingly, we have per-prompt metrics: $\text{Pass@K}^{(q)} = 1 - \frac{\binom{G-C}{K}}{\binom{G}{K}}$ and $\text{CoT-Pass@K}^{(q)} = 1 - \frac{\binom{G-D}{K}}{\binom{G}{K}}$. Besides, we estimate the probability of producing correct answers for prompt $q$ as $P(CA)^{(q)} = \frac{C}{G}$ and the probability of producing correct CoTs when generating correct answers as $P(CC|CA)^{(q)} = \frac{D}{C}$. For a dataset of multiple prompts, we take an average of per-prompt metrics as the dataset-level score.

**Optimization Effects**  In our reproduced DAPO training, we observe that most training questions have been fully optimized. As shown in Figure 4, the probability of generating correct answers for these questions almost reach 1. In the meanwhile, we also observe the improvement in producing more correct reasoning CoTs, as indicated by the improvements in $P(CC|CA)^{(q)}$. These observations validate the key perspective in Theorem 1: RLVR not only optimizes the final verifiable reward but also implicitly incentivizes correct reasoning.

**Generalization Behaviors**  In the meanwhile, Figure 5 discloses that the optimization of RLVR leads to the generalization improvements of both Pass@K and CoT-Pass@K from the very beginning. And using the CoT-Pass@K metric, we can clearly tell that the reasoning capability boundary has also been enhanced since the beginning. Another interpretation for this empirical observation

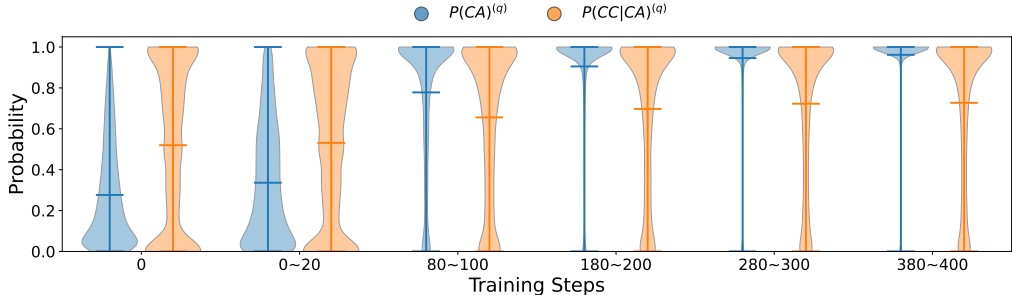

Figure 4: The evolution of $P(CA)^{(q)}$ (the fraction of correct answers for prompt $q$) and $P(CC|CA)^{(q)}$ (the fraction of correct CoTs within the correct answers for prompt $q$) for fully optimized training questions over the course of DAPO training.

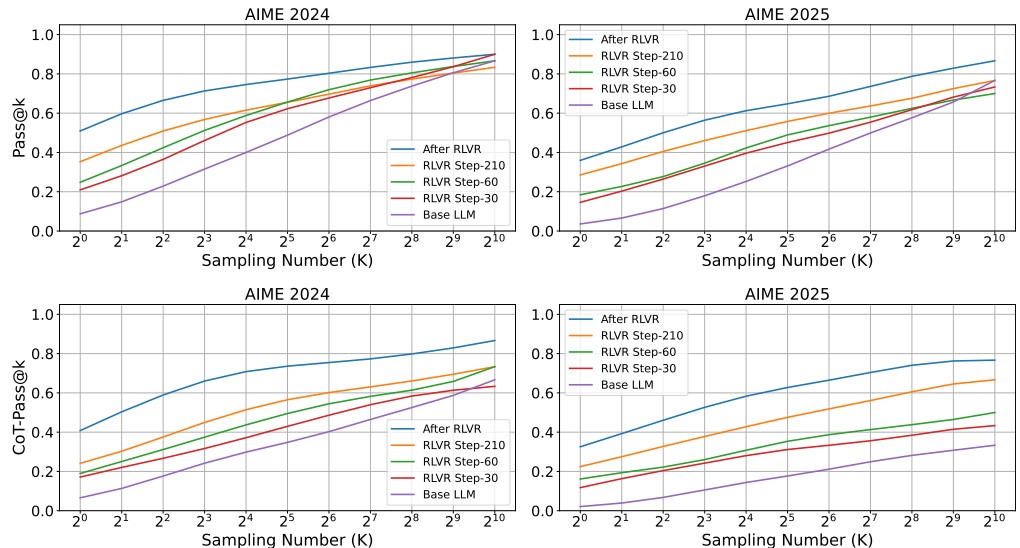

Figure 5: The evolution of Pass@K (the top row) and CoT-Pass@K (the bottom row) performance on AIME 2024 and 2025 for different model checkpoints during the DAPO training.

is that the model has learned to produce more and more reasoning CoTs that DeepSeek-R1-0528-Qwen-8B cannot identify any error. This ability is learned implicitly since we have no formal supervision for the CoT correctness.

**Limitations of DAPO** Figure 4 also unveils the limitations of DAPO, the R1-zero approach applied to a 32B base LLM. As $P(CA)^{(q)}$ approaches $1.0$ for most fully optimized training questions after 400 steps, which means these questions are no longer learnable because we cannot calculate a valid GRPO advantage for a all-correct group, we can still observe a non-negligible portion of imperfect CoTs (the median of $P(CC|CA)^{(q)}$ is around $0.7$). These signals indicate that there are certain unexpected reasoning behaviors learned in DAPO and we may not have a chance to mitigate them purely based on answer correctness as the reward.

## 6 THE QUALITY OF REASONING COTS ENHANCED BY RLVR

In addition to the LLM-as-a-CoT-Judge approach for strictly identifying critical errors in reasoning CoTs, we further leverage supervised fine-tuning (SFT) to assess the quality of reasoning CoTs enhanced by RLVR. Given the training questions in DAPO, we conduct multiple SFT procedures, starting from the same base LLM and learning from CoTs generated by different models. If the CoT

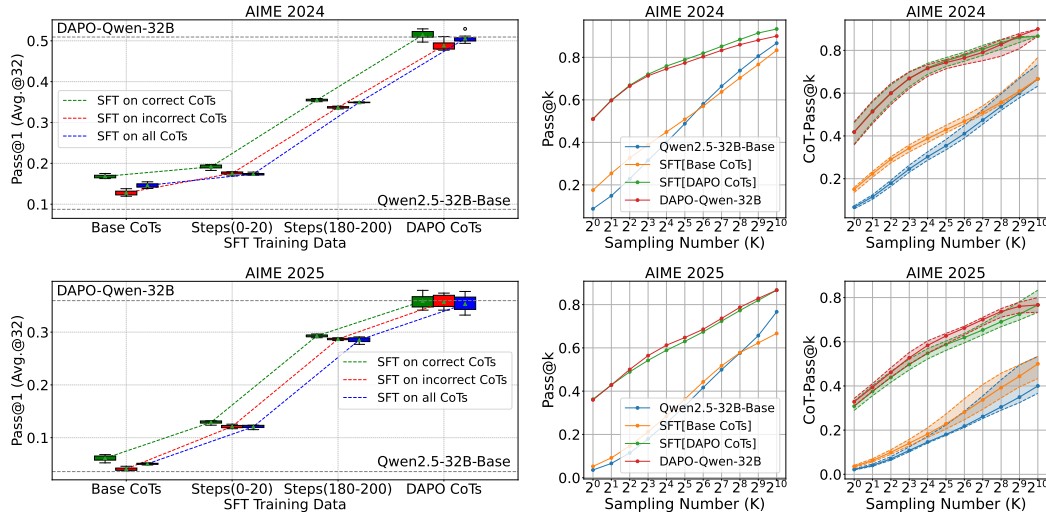

(a) The CoT quality at different RLVR stages, using Pass@1 on test sets as the proxy metric.

(b) The CoT quality before and after RLVR, using (CoT-)Pass@K on test sets as the proxy metric.

Figure 6: We show the generalization performance of post-SFT LLMs optimized on different CoT data. All these SFT processes start from the same base LLM, Qwen2.5-32B, with the only variable being the different CoT data on DAPO training questions. We use the performance on test sets (AIME 2024, 2025) as a proxy for the quality of the corresponding CoT data.

data is of high quality, we expect the post-SFT model to exhibit improved generalization performance. Figure 6 presents an overall quality evaluation of various CoT data.

Specifically, Figure 6(a) illustrates the evolution of CoT quality during RLVR. As training progresses, the generalization performance of post-SFT LLMs, measured in Pass@1, improves steadily. Ultimately, SFT on DAPO CoT data matches the Pass@1 performance of DAPO-Qwen-32B. This result indicates that, given sufficient training questions and CoT data from a post-RLVR model, we can replicate a new model with nearly the same Pass@1 performance simply through SFT. Moreover, an interesting observation is that, regardless of whether the CoT data contains identifiable errors, as RLVR progresses, the CoT quality, measured in Pass@1, generally improves. This suggests that although some erroneous steps may be present, the overall quality of these "incorrect" CoTs in the later stages of RLVR improves significantly.

Figure 6(b) compares the CoT quality before and after RLVR, using both Pass@K and CoT-Pass@K as proxy metrics. Comparing DAPO-Qwen-32B with the post-SFT model trained on its CoT data, we observe that this simple SFT approach nearly replicates the performance of a post-RLVR model, which would otherwise require significant computational cost. When comparing Qwen2.5-32B with the post-SFT model trained on its CoT data, we find that the post-SFT model begins to mitigate guessing. This aligns with our expectations, as we only feed CoT data with correct answers from Qwen2.5-32B into the SFT procedure. We can therefore regard this process as a round of off-policy RLVR optimization. These observations indicate that the incentivized CoT data through RLVR is crucial, as such CoTs cannot be directly sampled from base LLMs. RLVR optimizes the model's reasoning abilities, ensuring that the generated CoTs are more accurate, coherent, and reliable, which is essential for handling complex tasks.

**Limitations**  A key limitation of our study lies in the use of a LLM as the verifier for the correctness of reasoning CoTs, due to the prohibitive cost of manually checking a large volume of generated reasoning paths. Moreover, our theorem only explains the optimization process of RLVR but provides no guarantee for its generalization. We merely observe the generalization empirically. Due to space limitations, further discussions on the implications of our findings are deferred to Appendix A.6.

## 7 CONCLUSION

In this work, we address the fundamental problem of whether RLVR genuinely incentivizes novel reasoning in base LLMs. Through empirical evaluations and theoretical analysis, we justify a new perspective: RLVR implicitly incentivizes correct reasoning. Moreover, our analyses on training dynamics and CoT quality further confirm that the reasoning CoTs after RLVR are fundamentally different and can even help to replicate similar capabilities simply via supervised learning.

We hope these findings can not only resolve conflicting conclusions in prior work but also illuminate the untapped potential of RLVR in aligning LLMs with human reasoning systems. We envision a promising future where RLVR serves as a cornerstone for developing LLMs that learn through interaction, self-correction, and verifiable reasoning.

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

# A  APPENDIX

## A.1  DATA SOURCES

For math benchmarks studied in this paper, we leverage the following data sources: AIME 2025[1], AIME 2024[2], Math-500[3], AMC23[4], Minerva[5].

For training and evaluation of DAPO (Yu et al., 2025), we reuse their training data (`https://huggingface.co/datasets/BytedTsinghua-SIA/DAPO-Math-17k`) and processed version of AIME 2024 (`https://huggingface.co/datasets/BytedTsinghua-SIA/AIME-2024`). Please note that they have duplicated questions multiple times and explained in the dataset page that the purpose is to be compatible with an old version of VERL (Sheng et al., 2025). We reuse the prompt template of DAPO to evaluate their $Pass@K$ and $CoT\text{-}Pass@K$ performance on other benchmarks.

Besides, we follow the official LiveCodeBench repository[6] to perform Pass@K evaluations on competitive coding.

## A.2  LLM-AS-A-COT-JUDGE FOR MATH REASONING

We use a much more specialized LLM on mathematical reasoning (DeepSeek-R1-0528-Qwen3-8B) as the verifier to examine the reasoning steps of base LLM, Qwen2.5-32B. Meanwhile, we also acknowledge the existence of verification errors and manually checked many of them to confirm the reliability of this verification. To further mitigate potential verification errors, we design a multi-verification approach, as shown in Figure 7). For each reasoning CoT, we conduct multiple verifications independently and calculate three aggregation metrics:

- *All-correct*: Chains of Thought that pass all verification attempts
- *Majority-correct*: Chains of Thought that pass most verification attempts
- *Any-correct*: Chains of Thought that pass at least one verification attempt (capturing potential error recovery cases)

---

[1] `https://huggingface.co/datasets/opencompass/AIME2025`
[2] `https://huggingface.co/datasets/HuggingFaceH4/aime_2024`
[3] `https://huggingface.co/datasets/HuggingFaceH4/MATH-500`
[4] `https://huggingface.co/datasets/math-ai/amc23`
[5] `https://huggingface.co/datasets/math-ai/minervamath`
[6] `https://github.com/LiveCodeBench/LiveCodeBench`

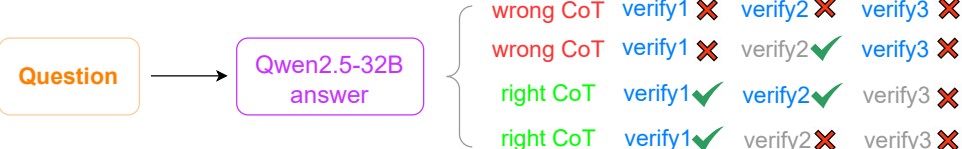

Figure 7: An intuitive diagram to illustrate the benefits of our multi-verification system: simultaneously considering *any-correct*, *all-correct*, and *majority-correct* helps us to mitigate false positives and false negatives within individual verifications.

This multi-verification approach can ensure us to have a comprehensive view of CoT correctness because the *all-correct* strategy mitigates false positives and the *any-correct* option reduces false negatives. Let $p_{\text{fp}}$ and $p_{\text{fn}}$ represent the per-attempt false positive and false negative rates, respectively. For $n$ independent verification attempts, we observe:

- *All-correct*: The false positive rate decays exponentially as $p_{\text{fp}}^n$
- *Any-correct*: The false negative rate decays exponentially as $p_{\text{fn}}^n$

In our study, we employ $n = 3$ verification attempts for each CoT.

Moreover, we provide the prompt template used for DeepSeek-R1-0528-Qwen3-8B as follows.

---

**Our Prompt Template for Verifier DeepSeek-R1-0528-Qwen3-8B**

You are an expert in mathematics and logical reasoning. Your task is to evaluate the correctness of a solution to a given math problem, with a **strong emphasis on the reasoning process**, not just the final answer.
Below is the **Problem** and the **Solution (Provided by another AI model)**:
—

**Problem**:
{{question}}
**Solution (Provided by another AI model)**:
{{solution}}
—

Please perform the following tasks:
1. **Analyze the solution step-by-step**, paying close attention to: - Computational accuracy - Logical consistency - Conceptual understanding - Whether the reasoning is valid and complete
2. **Identify any issues or errors in the reasoning**, even if the final answer is correct. Classify them into the following categories (if applicable): - **Calculation Error**: Mistakes in arithmetic, algebraic manipulation, or numerical computation. - **Logical Error**: Invalid reasoning, flawed logic, or incorrect inference. - **Conceptual Error**: Misunderstanding or misuse of mathematical concepts or definitions. - **Omission / Incompleteness**: Missing steps, incomplete justification, or not addressing all parts of the question. - **Other**: Any other type of error that does not fit into the above categories.
3. **Provide a final judgment** on whether the solution is logically sound and free of errors in reasoning.
Please format your response as follows:
—

**Issues Identified:**
- [Issue 1]: [Classification] - [Brief explanation] - [Issue 2]: [Classification] - [Brief explanation] - ...
Let's think step by step and output your final judgment within \boxed{}
\boxed{yes} or \boxed{no}

### A.3 REVISITING PASS@K EXPERIMENTS FOR SKYWORK-OR1

Skywork-OR1 (He et al., 2025) has generously shared their complete training recipes, claiming to enhance distilled LLMs with more powerful reasoning capabilities through RLVR. Therefore, we conduct the Pass@K experiments on their models to understand how RLVR could improve distilled LLMs.

Figure 8 shows the Pass@K comparisons between Skywork-OR1-7B and DeepSeek-R1-Distill-Qwen-7B on LiveCodeBench-v6. We can also observe a significant improvement of both sampling efficiency (Pass@1) and reasoning boundary (Pass@K, K up to 1024). This conclusion is consistent with the observations for Figure 3 in the main paper.

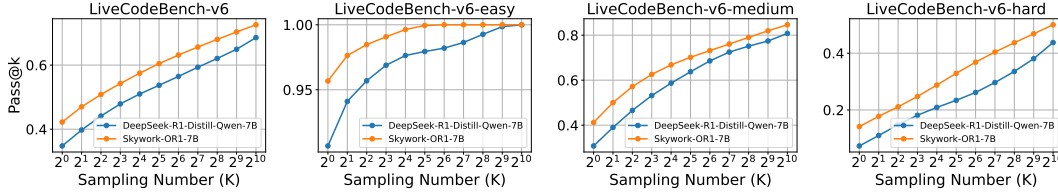

Figure 8: Comparisons of Pass@K on LiveCodeBench-v6 and its different difficulty-level subsets. Here the distilled LLM is DeepSeek-R1-Distill-Qwen-7B, and the post-RLVR model is Skywork-OR1-7B.

However, in math domains, applying RLVR to distilled LLMs seems to merely deliver sampling efficiency improvements. As shown in Figure 9, we observe that even using the CoT-Pass@K metric, Skywork-OR1-Math-7B and DeepSeek-R1-Distill-Qwen-7B do not have distinct Pass@K gaps for large K values. We suspect the reason is that the distilled LLM may already master major reasoning capabilities that can be learned with RLVR using answer correctness as the reward. So in math domains, their main improvements lie in Pass@1. In contrast, for code domains, applying RLVR to distilled LLMs can still stimulate them to fit for real-world execution feedback, thereby incentivizing extended reasoning boundary.

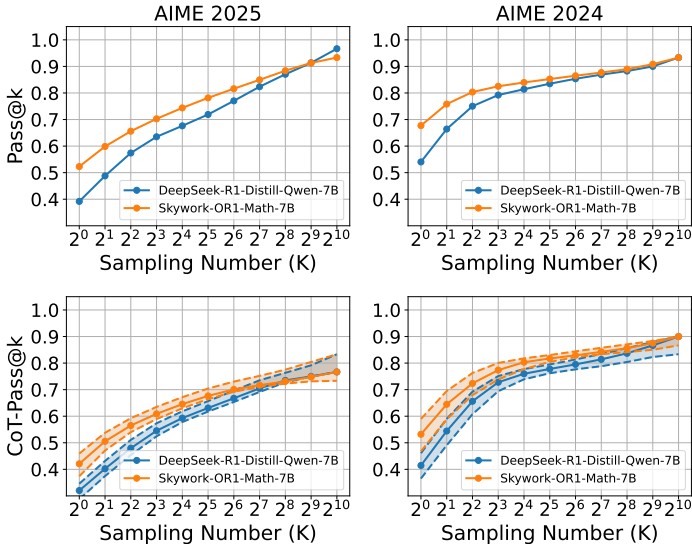

Figure 9: Comparisons of Pass@K (the top row) and CoT-Pass@K (the bottom row) on AIME 2024, 2025 to show how RLVR could improve distilled LLMs. Here the distilled LLM is DeepSeek-R1-Distill-Qwen-7B, and the post-RLVR model is Skywork-OR1-Math-7B. For CoT-Pass@K, we perform multiple verifications for each CoT using DeepSeek-R1-0528-Qwen3-8B, and display the results determined by *any-correct*, *all-correct*, and *majority-correct* strategies, which constitute the shaded area in lower subplots.

A.4 PROOF AND ADDITIONAL THEORETICAL ANALYSIS

Below we include the detailed proof for Theorem 1.

**Proof 1** *Let $p_c = P(\mathcal{I}_{CoT}(c_i) = 1)$ be the current probability of generating a correct CoT. The expected reward for a response $y_i$ is:*

$$\mathbb{E}[R(y_i)] = \begin{cases} \alpha & \text{if } \mathcal{I}_{CoT}(c_i) = 1 \\ \beta & \text{if } \mathcal{I}_{CoT}(c_i) = 0 \end{cases} \tag{6}$$

*The group-level expected reward $\mu \triangleq \mathbb{E}[\mu_\mathbf{Y}]$ is:*

$$\mu = p_c\alpha + (1 - p_c)\beta. \tag{7}$$

*For large $G$, the group mean $\mu_\mathbf{Y}$ and variance $\sigma_\mathbf{Y}^2$ concentrate around their expectations:*

$$\mu_\mathbf{Y} \xrightarrow{G \to \infty} \mu \tag{8}$$

$$\sigma_\mathbf{Y}^2 \xrightarrow{G \to \infty} \sigma^2 > 0. \tag{9}$$

*The expected advantage conditional on CoT correctness is:*

$$\mathbb{E}[\hat{A}(y_i) \mid \mathcal{I}_{CoT}(c_i) = 1] \xrightarrow{G \to \infty} \frac{\alpha - \mu}{\sigma} \tag{10}$$

$$\mathbb{E}[\hat{A}(y_i) \mid \mathcal{I}_{CoT}(c_i) = 0] \xrightarrow{G \to \infty} \frac{\beta - \mu}{\sigma}. \tag{11}$$

*Substituting equation 7 into equation 10 and equation 11:*

$$\mathbb{E}[\hat{A}(y_i) \mid \text{correct CoT}] \to \frac{(1 - p_c)(\alpha - \beta)}{\sigma} \tag{12}$$

$$\mathbb{E}[\hat{A}(y_i) \mid \text{incorrect CoT}] \to \frac{-p_c(\alpha - \beta)}{\sigma}. \tag{13}$$

*Since $\alpha > \beta$ (by equation 4 under the* Logic Prior *assumption) and $\sigma > 0$, we have:*

$$(1 - p_c)(\alpha - \beta)/\sigma > 0,$$
$$-p_c(\alpha - \beta)/\sigma < 0,$$

*proving inequalities equation 5.*

*The GRPO policy gradient update in equation 3, $\nabla_\theta J(\theta) \approx \frac{1}{G} \sum_{i=1}^{G} \hat{A}(y_i)\nabla_\theta \log \pi_\theta(y_i \mid q)$, on average increases the likelihood of responses with $\hat{A}(y_i) > 0$ (correct CoTs) and decreases it for $\hat{A}(y_i) < 0$ (incorrect CoTs). Thus, $p_c$ increases monotonically.*

**Discussions on $(\mu, \sigma^2)$ in Theorem 1** From equation 7, we know that the group reward mean is given by $\mu = p_c\alpha + (1 - p_c)\beta$. Furthermore, we can derive the exact formula for the variance $\sigma^2$ in equation 9 and analyze their impacts together with $p_c$, $\alpha$, and $\beta$ on policy iterations.

The sample variance $\sigma_\mathbf{Y}^2$ converges to the true variance $\sigma^2$:

$$\sigma_\mathbf{Y}^2 = \frac{1}{G} \sum_{j=1}^{G} (R(y_j) - \mu_\mathbf{Y})^2 \xrightarrow{G \to \infty} \text{Var}(R(y_j)) \equiv \sigma^2,$$

where $\text{Var}(R(y_j))$ can be computed using the law of total variance:

$$\text{Var}(R(y_j)) = \underbrace{\text{Var}(\mathbb{E}[R(y_j) \mid \mathcal{I}_{\text{CoT}}(c_j)])}_{\text{Variance of conditional expectation}} + \underbrace{\mathbb{E}[\text{Var}(R(y_j) \mid \mathcal{I}_{\text{CoT}}(c_j))]}_{\text{Expectation of conditional variance}}.$$

First term:

$$\mathbb{E}[R(y_j) \mid \mathcal{I}_{\text{CoT}}(c_j)] = \begin{cases} \alpha & \text{if } \mathcal{I}_{\text{CoT}}(c_j) = 1 \\ \beta & \text{if } \mathcal{I}_{\text{CoT}}(c_j) = 0 \end{cases}.$$

The random variable $\mathbb{E}[R(y_j) \mid \mathcal{I}_{\text{CoT}}(c_j)]$ has variance:

$$\text{Var}(\mathbb{E}[R(y_j) \mid \mathcal{I}_{\text{CoT}}(c_j)]) = (\alpha - \beta)^2 p_c (1 - p_c).$$

Second term:

$$\text{Var}(R(y_j) \mid \mathcal{I}_{\text{CoT}}(c_j)) = \begin{cases} \alpha(1 - \alpha) & \text{if } \mathcal{I}_{\text{CoT}}(c_j) = 1 \\ \beta(1 - \beta) & \text{if } \mathcal{I}_{\text{CoT}}(c_j) = 0 \end{cases},$$

so its expectation is:

$$\mathbb{E}[\text{Var}(R(y_j) \mid \mathcal{I}_{\text{CoT}}(c_j))] = p_c \alpha(1 - \alpha) + (1 - p_c)\beta(1 - \beta).$$

Thus:

$$\sigma^2 = (\alpha - \beta)^2 p_c(1 - p_c) + p_c \alpha(1 - \alpha) + (1 - p_c)\beta(1 - \beta). \tag{14}$$

Substituting $\mu$ and $\sigma$ into equation 12 and equation 13, we have

$$\mathbb{E}[\hat{A}(y_i) \mid \text{correct CoT}] \to \frac{(1 - p_c)(\alpha - \beta)}{\sqrt{(\alpha - \beta)^2 p_c(1 - p_c) + p_c \alpha(1 - \alpha) + (1 - p_c)\beta(1 - \beta)}},$$

$$\mathbb{E}[\hat{A}(y_i) \mid \text{incorrect CoT}] \to \frac{-p_c(\alpha - \beta)}{\sqrt{(\alpha - \beta)^2 p_c(1 - p_c) + p_c \alpha(1 - \alpha) + (1 - p_c)\beta(1 - \beta)}}.$$

An ideal pre-training on a high-capacity model could help to ensure that $\alpha \to 1$ and $\beta \to 0$ at the beginning of RLVR. In this condition, we have the following advantage estimates:

$$\mathbb{E}[\hat{A}(y_i) \mid \text{correct CoT}] \to \sqrt{\frac{1 - p_c}{p_c}}, \quad \mathbb{E}[\hat{A}(y_i) \mid \text{incorrect CoT}] \to -\sqrt{\frac{p_c}{1 - p_c}}.$$

In this ideal scenario, the role of human would be to prepare a comprehensive and diverse set of questions and answers, leveraging RLVR to automatically incentivize the model's reasoning capabilities. However, in practice—the "unideal case"—it is often necessary to first fine-tune the base LLM to align its output with a proper reasoning distribution before applying RLVR.

**Discussions on Key Observations in RLVR** Grounded in our theoretical analysis, we can now provide our unique explanations for several previously elusive yet important observations reported in DeepSeek-R1 (Guo et al., 2025).

Our Explanation of the Observation "*DeepSeek-R1-Zero achieved remarkable Pass@K performance on AIME 2024 but encountered challenges such as poor readability and language mixing.*": Even DeepSeek-V3 (Liu et al., 2024) cannot guarantee ideal conditions where $\alpha \to 1, \beta \to 0$. As a result, cold-start data is required to rectify prior logic biases, motivating the R1 approach.

Our Explanation of the Observation "*The R1-Zero approach did not work well for the 32B dense model, yet distillation can be very effective.*": Key factors such as $(p_c, \alpha, \beta)$ for the 32B base model are in an even worse state, causing pure RLVR to converge to suboptimal local solutions. Based on our analysis, the key to effective reasoning lies in learning correct CoTs. Therefore, the distillation approach can efficiently teach an LLM how to reason properly.

## A.5 ADDITIONAL DETAILS IN REPRODUCING DAPO TRAINING

Our reproduction was conducted on 32 AMD MI300X GPUs using the VERL framework (Sheng et al., 2025), and ran for over two weeks. Our run did not fully reproduce the $Pass@1$ accuracy above 50% as reported by Yu et al. (2025), while we reached a comparable performance of around 44% $Pass@1$, in line with a third-party reproduction (Chen et al., 2025a). We use the same verifier introduced in Section 3 to assess the correctness of both training and evaluation rollouts.

In addition to the performance evolution on fully optimized training questions highlighted in Figure 4 of the main paper, we include performance evolution on hard questions and more continuous validation performance in Figure 10 to provide more comprehensive information. The additional observations are consistent with what we have introduced in the main paper. It is natural to observe that RLVR begins to incentivize correct reasoning from the very beginning, as evidenced by

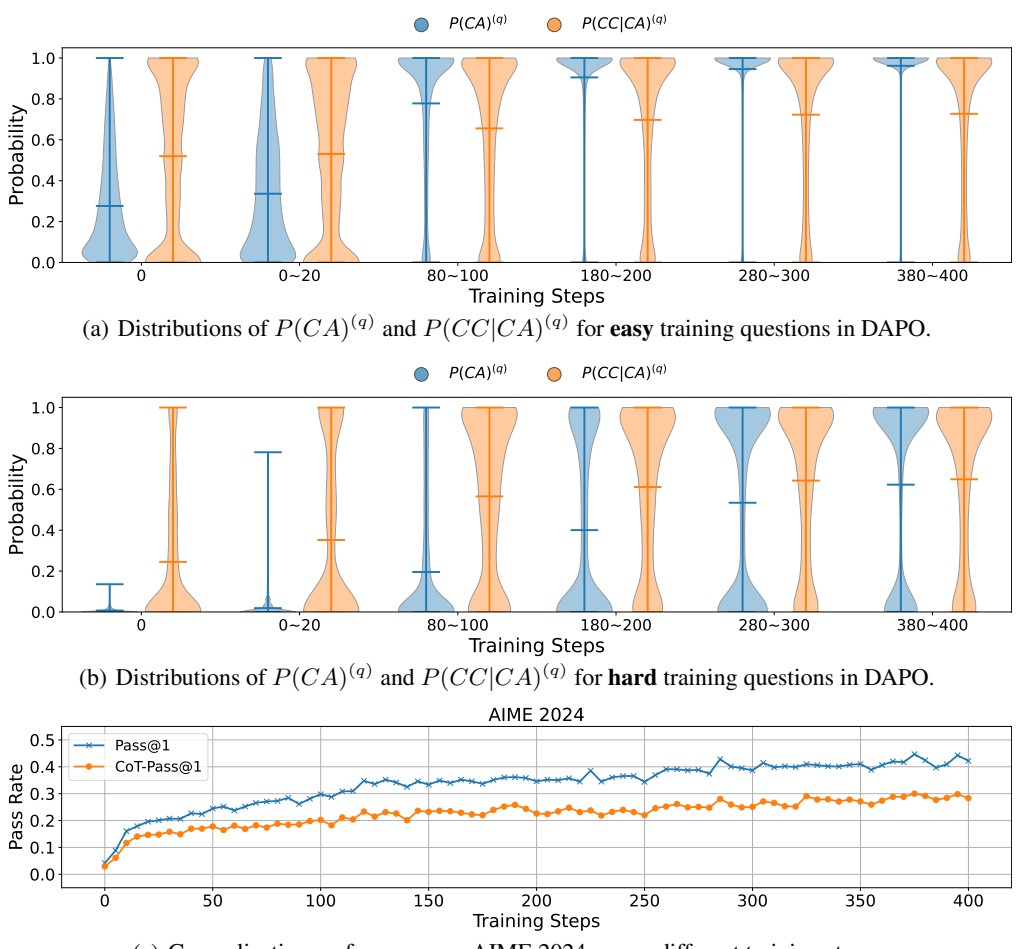

(a) Distributions of $P(CA)^{(q)}$ and $P(CC|CA)^{(q)}$ for **easy** training questions in DAPO.

(b) Distributions of $P(CA)^{(q)}$ and $P(CC|CA)^{(q)}$ for **hard** training questions in DAPO.

(c) Generalization performance on AIME 2024 across different training steps.

Figure 10: We show gradually optimized performance on training prompts throughout RLVR and corresponding generalization behaviors on testing prompts. The top two subfigures use violin plots to visualize the evolution of $P(CA)^{(q)}$ (the fraction of correct answers for prompt $q$) and $P(CC|CA)^{(q)}$ (the fraction of correct CoTs within the correct answers for prompt $q$) over the course of DAPO training. Subfigure (a) shows results for "easy" training questions, while (b) presents the corresponding distributions for "hard" questions ($P(CA)^{(q)}$ ¡ 1 for DAPO). We divide training questions into easy and hard by sampling 64 rollouts for each of the $17k$ training questions using Qwen2.5-32B, and labeling those with at least one correct answer as easy. Subfigure (c) presents the generalization performance on AIME 2024 across different training steps.

increased $P(CC|CA)^{(q)}$ values in the early training steps shown in Figures 10(a) and 10(b). These incentivized reasoning capabilities translate into improved generalization on unseen questions, as demonstrated by notable gains in CoT-Pass@K on AIME 2024 within the first 20 training steps in Figure 10(c). Note that each training step here corresponds to one round of PPO-style optimization (Schulman et al., 2017), which includes 16 gradient updates, according to the DAPO training script. Thus, we see that correct reasoning abilities begin to generalize after only a few gradient updates.

Furthermore, the incentivization of correct reasoning on training questions appears to be a continuous process, as reflected by the steady increase in the mean of $P(CC|CA)^{(q)}$ throughout training, for both easy and hard questions. Meanwhile, we again observe that $P(CA)^{(q)}$ (equivalent to $Pass@1^{(q)}$) is an unreliable metric, particularly for easy training questions. As shown in Figure 10(a), the distribution of $P(CA)^{(q)}$ becomes highly skewed toward 1.0 after 180 steps, mislead-

ingly suggesting that most questions are perfectly solved. However, examining the distribution of $P(CC|CA)^{(q)}$ reveals that a substantial fraction of responses still contain flawed reasoning. We suspect this is one of the reasons behind the difficulty of achieving strong results with Qwen2.5-32B using the R1-zero approach.

For both easy and hard training questions, improving $P(CC|CA)^{(q)}$ seems to be a slow and challenging process. Since our analysis shows that enhancing correct CoTs is key to improving reasoning capabilities, we believe that future research should explore novel mechanisms to accelerate the improvement of $P(CC|CA)^{(q)}$, thereby enhancing both the efficiency and effectiveness of RLVR.

## A.6 DISCUSSIONS

**Call for Live, Challenging Benchmarks**   Static benchmarks developed prior to the release of modern base models are increasingly susceptible to contamination risks, potentially undermining the reliability of observed improvements. In response, we emphasize the need for *live benchmarks* that evolve over time, as suggested in recent studies (Jain et al., 2025; White et al., 2025). Additionally, we agree with the viewpoint of Yao (2025) that future research advancements may rely more on designing new evaluations, benchmarks, and environments.

**Call for Lightweight yet Powerful CoT Verifiers**   While DeepSeek-R1-0528-Qwen3-8B serves as a useful CoT verifier, it is not infallible. Conflicting verification results across multiple queries reveal the challenges of false-positive and false-negative verifications. To tackle this, we combine multiple verification strategies, including different voting rules, to improve robustness. Looking forward, there is a pressing need for light yet reliable CoT verifiers that can serve as standardized evaluators beyond the coarse-grained Pass@K metric. This direction also relates to previous studies on process reward modeling (Lightman et al., 2024; Uesato et al., 2022; Wang et al., 2024).

**Scaling RLVR or Scaling Pre-Training**   While the scaling of pre-training has led to transformative progress in LLMs (Kaplan et al., 2020; Liu et al., 2024), enabling the transition to the era of artificial general intelligence, we argue that scaling RLVR could be equally pivotal, given the empirical evidences and theoretical foundation that all demonstrate its real incentivization beyond base LLMs. As modern LLMs approach the limits of language token exposure, learning from experience (Silver & Sutton, 2025) may represent the next leap. Recent efforts by leading research teams suggest a growing emphasis on this direction (Guo et al., 2025; DeepSeek, 2025; Gemini, 2024; Grok, 2025; OpenAI, 2025; Qwen, 2025; Gemini, 2025; Anthropic, 2025; Mistral.AI, 2025). For the broad open research community, understanding the foundations and limitations of current RLVR algorithms is crucial to push this direction further.

**New RLVR Algorithms and Beyond**   With our insight that RLVR implicitly incentivizes correct reasoning in base LLMs, we anticipate the development of new algorithmic paradigms. These may include optimization formulations or objective functions, such as policy-gradient approaches (Sutton et al., 1999; Schulman et al., 2017), new likelihood-based optimization objectives (Chen et al., 2025a; Zhu et al., 2025), and preference optimization frameworks (Rafailov et al., 2023; Su et al., 2025). The key principle is that the new algorithms should be designed to more directly incentivize correct reasoning paths, alleviating inherent logical biases in base LLMs.

## A.7 Case Studies

We perform manual verification on difficult problems from multiple benchmarks, focusing on cases where the base model occasionally produces a correct final answer despite flawed reasoning in its CoT. For each case study, we present the original question, the full response generated by Qwen2.5-32B, and a detailed breakdown of key errors identified by DeepSeek-R1-0528-Qwen3-8B, explicitly linking each error to the corresponding flawed reasoning segments in the CoT. For layout and color schemes, we follow the visualization strategies used in Yue et al. (2025).

### A.7.1 Case Studies on AIME 2024

---

**Question**

**AIME24 I Question No.9 (dataset id 74)**[7]
Let $A$, $B$, $C$, and $D$ be points on the hyperbola $\frac{x^2}{20} - \frac{y^2}{24} = 1$ such that $ABCD$ is a rhombus whose diagonals intersect at the origin. Find the greatest real number that is less than $BD^2$ for all such rhombi.

---

**Qwen2.5-32B Answer**

The diagonals of a rhombus are perpendicular and bisect each other, so $ABCD$ is a square. Let $a = BD = AC$.

> DeepSeek-R1-0528-Qwen3-8B verify: - **Conceptual Error**: The solution incorrectly assumes that ABCD is a square, whereas the problem only specifies that it is a rhombus. A rhombus does not necessarily have equal diagonals, so this assumption is invalid.

Then $a^2 = 2(AB^2)$. The equation of the hyperbola can be rewritten as $y^2 = 24\left(\frac{x^2}{20} - 1\right) = \frac{6}{5}x^2 - 24$. Let $P$ be the point where the diagonals intersect. Then $OP = \frac{a}{2}$, so $P = \left(\frac{a}{2}, \frac{a}{2}\right)$.

> DeepSeek-R1-0528-Qwen3-8B verify: - **Logical Error**: The solution defines P as the intersection point and sets it to (a/2, a/2), but the problem states that the diagonals intersect at the origin. This inconsistency leads to incorrect reasoning, as P should be (0,0) based on the problem statement.

Since $P$ lies on the hyperbola, we have $\frac{a^2}{4} = \frac{6}{5}\left(\frac{a^2}{4}\right) - 24$, so $\frac{a^2}{20} = 24$, or $a^2 = 480$.

Therefore, the greatest real number that is less than $BD^2$ for all such rhombi is $\boxed{480}$.

---

**Question**

**AIME24 II Question No.15(dataset id 81)**[8]
Find the number of rectangles that can be formed inside a fixed regular dodecagon (12-gon) where each side of the rectangle lies on either a side or a diagonal of the dodecagon. The diagram below shows three of those rectangles.

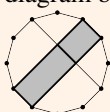

---

[7]https://artofproblemsolving.com/wiki/index.php/2024_AIME_I_Problems/Problem_9

[8]https://artofproblemsolving.com/wiki/index.php/2024_AIME_II_Problems/Problem_15

**Qwen2.5-32B Answer**

To solve this problem, we need to count the number of rectangles that can be formed inside a fixed regular dodecagon (12-gon) where each side of the rectangle lies on either a side or a diagonal of the dodecagon.

First, let's consider the sides of the rectangle. Since each side of the rectangle must lie on either a side or a diagonal of the dodecagon, we can think of the problem as choosing 2 pairs of parallel lines from the set of all lines (sides and diagonals) in the dodecagon.

There are 12 sides in a dodecagon. Additionally, there are 12 diagonals that connect every other vertex, and 12 diagonals that connect every third vertex. So, there are a total of 36 lines in the dodecagon.

Now, let's count the number of ways to choose 2 pairs of parallel lines. We can do this by considering the different types of lines and the number of ways to choose 2 lines from each type:

1. Choosing 2 lines from the 12 sides: There are 12 choose 2 ways to do this, which is equal to 66.

> DeepSeek-R1-0528-Qwen3-8B verify: the method of counting two pairs of parallel lines assumes that any such choice forms a rectangle, which is not necessarily true as the lines may not intersect properly to form a rectangle.

2. Choosing 2 lines from the 12 diagonals that connect every other vertex: There are 12 choose 2 ways to do this, which is equal to 66.
3. Choosing 2 lines from the 12 diagonals that connect every third vertex: There are 12 choose 2 ways to do this, which is equal to 66.
4. Choosing 1 line from the sides and 1 line from the diagonals that connect every other vertex: There are 12 * 12 ways to do this, which is equal to 144.
5. Choosing 1 line from the sides and 1 line from the diagonals that connect every third vertex: There are 12 * 12 ways to do this, which is equal to 144.
6. Choosing 1 line from the diagonals that connect every other vertex and 1 line from the diagonals that connect every third vertex: There are 12 * 12 ways to do this, which is equal to 144.

> DeepSeek-R1-0528-Qwen3-8B verify: **Logical Error**: The solution only considers three types of lines (sides, diagonals connecting every other vertex, and diagonals connecting every third vertex), but there are additional diagonals (steps 4, 5, and 6). By ignoring these, the classification is incomplete.

Adding up all these possibilities, we get a total of 66 + 66 + 66 + 144 + 144 + 144 = 630. However, we have counted each rectangle twice (once for each pair of parallel lines), so we need to divide the total by 2 to get the final answer: 630 / 2 = 315.

Therefore, the number of rectangles that can be formed inside a fixed regular dodecagon (12-gon) where each side of the rectangle lies on either a side or a diagonal of the dodecagon is $\boxed{315}$.

**Question**

**AIME24 I Question No.11(dataset id 62)**[9]
Each vertex of a regular octagon is independently colored either red or blue with equal probability. The probability that the octagon can then be rotated so that all of the blue vertices end up at positions where there had been red vertices is $\frac{m}{n}$, where $m$ and $n$ are relatively prime positive integers. Find $m + n$.

---

[9]https://artofproblemsolving.com/wiki/index.php/2024_AIME_I_Problems/Problem_11

**Qwen2.5-32B Answer**

To solve this problem, we need to consider the possible colorings of the vertices of the octagon and how they can be rotated.

First, let's consider the number of ways to color the vertices of the octagon. Since each vertex can be colored either red or blue, there are $2^8 = 256$ possible colorings.

Next, we need to consider the rotations of the octagon. Since the octagon has 8 vertices, there are 8 possible rotations.

Now, let's consider the condition that all blue vertices end up at positions where there were originally red vertices. This means that the number of blue vertices must be less than or equal to the number of red vertices. Since there are 8 vertices in total, the number of blue vertices can be 0, 1, 2, 3, or 4.

For each of these cases, we need to count the number of colorings that satisfy the condition and divide by the total number of colorings to find the probability.

Case 1: 0 blue vertices There is only 1 way to color the vertices in this case (all red), and it satisfies the condition. The probability is $\frac{1}{256}$.

Case 2: 1 blue vertex There are $\binom{8}{1} = 8$ ways to choose the position of the blue vertex, and all of them satisfy the condition. The probability is $\frac{8}{256} = \frac{1}{32}$.

Case 3: 2 blue vertices There are $\binom{8}{2} = 28$ ways to choose the positions of the blue vertices. However, not all of them satisfy the condition. We need to subtract the cases where the blue vertices are adjacent. There are 8 such cases (one for each rotation of the octagon). So, there are $28 - 8 = 20$ colorings that satisfy the condition. The probability is $\frac{20}{256} = \frac{5}{64}$.

> DeepSeek-R1-0528-Qwen3-8B verify: - **Logical Error**: The solution incorrectly assumes that subtracting cases based on adjacency or specific shapes (e.g., "V" shape) ensures the condition does not hold. However, the condition depends on the existence of a rotation that maps all blue vertices to red positions, not on the adjacency or shape of the blue vertices. For example, with two blue vertices, adjacent configurations can still satisfy the condition for some rotations, so subtracting them is invalid.

Case 4: 3 blue vertices There are $\binom{8}{3} = 56$ ways to choose the positions of the blue vertices. However, not all of them satisfy the condition. We need to subtract the cases where the blue vertices are adjacent or form a "V" shape. There are 16 such cases (one for each rotation of the octagon). So, there are $56 - 16 = 40$ colorings that satisfy the condition. The probability is $\frac{40}{256} = \frac{5}{32}$.

Case 5: 4 blue vertices There are $\binom{8}{4} = 70$ ways to choose the positions of the blue vertices. However, not all of them satisfy the condition. We need to subtract the cases where the blue vertices are adjacent or form a "V" shape. There are 24 such cases (one for each rotation of the octagon). So, there are $70 - 24 = 46$ colorings that satisfy the condition. The probability is $\frac{46}{256} = \frac{23}{128}$.

Now, we need to sum up the probabilities for all cases and simplify the fraction:

$$\frac{1}{256} + \frac{1}{32} + \frac{5}{64} + \frac{5}{32} + \frac{23}{128} = \frac{1 + 8 + 20 + 40 + 46}{256} = \frac{115}{256}$$

The fraction $\frac{115}{256}$ is already in its simplest form, so $m = 115$ and $n = 256$. Therefore, $m + n = 115 + 256 = \boxed{371}$.

---

**Question**

**AIME24 I Question No.14(dataset id 65)**[10]

Let $ABCD$ be a tetrahedron such that $AB = CD = \sqrt{41}$, $AC = BD = \sqrt{80}$, and $BC = AD = \sqrt{89}$. There exists a point $I$ inside the tetrahedron such that the distances from $I$ to each of the faces of the tetrahedron are all equal. This distance can be written in the form $\frac{m\sqrt{n}}{p}$, where $m$, $n$, and $p$ are positive integers, $m$ and $p$ are relatively prime, and $n$ is not divisible by the square of any prime. Find $m + n + p$.

---

**Qwen2.5-32B Answer**

To find the distance from the point $I$ to each of the faces of the tetrahedron, we can use the formula for the volume of a tetrahedron and the fact that the distances from $I$ to each of the faces are equal.

First, let's find the volume of the tetrahedron using Heron's formula for the area of a triangle and the formula for the volume of a tetrahedron:

$V = \frac{1}{6} \cdot$ base area $\cdot$ height

We can choose any face as the base, so let's use the face $ABC$:

$AB = \sqrt{41}$, $AC = \sqrt{80}$, and $BC = \sqrt{89}$

Using Heron's formula, we can find the area of the triangle $ABC$:

$s = \frac{AB+AC+BC}{2} = \frac{\sqrt{41}+\sqrt{80}+\sqrt{89}}{2}$

$A_{ABC} = \sqrt{s(s-AB)(s-AC)(s-BC)}$

Now, let's find the height of the tetrahedron from the point $I$ to the face $ABC$. We know that the distances from $I$ to each of the faces are equal, so let's call this distance $h$.

$V = \frac{1}{6} \cdot A_{ABC} \cdot h$

> DeepSeek-R1-0528-Qwen3-8B verify: **Conceptual Error** - The volume formula $V = \frac{1}{6} \cdot A_{ABC} \cdot h$ is incorrect for the incenter of the tetrahedron

We can find the volume of the tetrahedron using the Cayley-Menger determinant:

$$V^2 = \frac{1}{288} \cdot \begin{vmatrix} 0 & 1 & 1 & 1 & 1 \\ 1 & 0 & 41 & 80 & 89 \\ 1 & 41 & 0 & 89 & 41 \\ 1 & 80 & 89 & 0 & 80 \\ 1 & 89 & 41 & 80 & 0 \end{vmatrix}$$

> DeepSeek-R1-0528-Qwen3-8B verify: - **Calculation Error** - The Cayley-Menger determinant matrix contains multiple incorrect values. For example, the element in row 2, column 5 should be 80 (BD²) but is 41, and the element in row 3, column 4 should be 41 (CD²) but is 80. This affects the volume calculation.

Solving for $V$, we get:

$V = \frac{1}{6} \cdot A_{ABC} \cdot h$

Now, we can solve for $h$:

$h = \frac{6V}{A_{ABC}}$

After calculating the values, we find that $h$ can be written in the form $\frac{m\sqrt{n}}{p}$, where $m$, $n$, and $p$ are positive integers, $m$ and $p$ are relatively prime, and $n$ is not divisible by the square of any prime.

The final answer is $m + n + p = \boxed{104}$.

---

[10] https://artofproblemsolving.com/wiki/index.php/2024_AIME_I_Problems/Problem_14

### A.7.2 CASE STUDIES ON AIME 2025

---

**Question**

**AIME25 I Question No.2** [11]
On $\triangle ABC$ points $A, D, E,$ and $B$ lie that order on side $\overline{AB}$ with $AD = 4, DE = 16,$ and $EB = 8$. Points $A, F, G,$ and $C$ lie in that order on side $\overline{AC}$ with $AF = 13, FG = 52,$ and $GC = 26$. Let $M$ be the reflection of $D$ through $F$, and let $N$ be the reflection of $G$ through $E$. Quadrilateral $DEGF$ has area 288. Find the area of heptagon $AFNBCEM$.

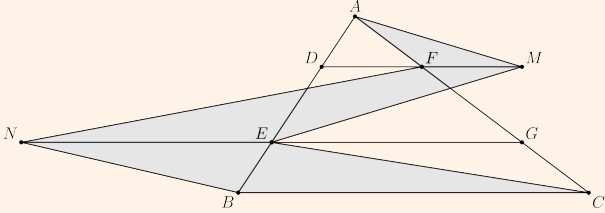

---

**Qwen2.5-32B Answer**

To solve the problem, we need to find the area of the heptagon $AFNBCEM$. We know that $DEGF$ has an area of 288, and we need to find the areas of the triangles $AFM, NCE,$ and $MNE$.

First, let's find the area of triangle $AFM$. Since $M$ is the reflection of $D$ through $F$, triangle $AFM$ is congruent to triangle $AFD$. The base of triangle $AFD$ is $AF = 13$, and its height is the same as the height of triangle $ADE$, which is $\frac{288}{DE+EG} = \frac{288}{16+52} = \frac{288}{68} = \frac{72}{17}$. So, the area of triangle $AFM$ is $\frac{1}{2} \cdot AF \cdot \text{height} = \frac{1}{2} \cdot 13 \cdot \frac{72}{17} = \frac{468}{17}$.

> DeepSeek-R1-0528-Qwen3-8B verify: - - **Conceptual Error**: The solution incorrectly assumes that the height of triangle ADE is derived from the area of quadrilateral DEGF using DE + EG, but EG is not defined in the problem.

Next, let's find the area of triangle $NCE$. Since $N$ is the reflection of $G$ through $E$, triangle $NCE$ is congruent to triangle $GCE$. The base of triangle $GCE$ is $GC = 26$, and its height is the same as the height of triangle $GEB$, which is $\frac{288}{GE+EB} = \frac{288}{52+8} = \frac{288}{60} = \frac{24}{5}$. So, the area of triangle $NCE$ is $\frac{1}{2} \cdot GC \cdot \text{height} = \frac{1}{2} \cdot 26 \cdot \frac{24}{5} = \frac{312}{5}$.

> DeepSeek-R1-0528-Qwen3-8B verify: Similarly, for triangle NCE, the height is calculated using GE + EB, but GE is not given; the solution uses 52 (FG) incorrectly.

Finally, let's find the area of triangle $MNE$. Since $M$ is the reflection of $D$ through $F$ and $N$ is the reflection of $G$ through $E$, triangle $MNE$ is congruent to triangle $DGE$. The base of triangle $DGE$ is $DE + EG = 16 + 52 = 68$, and its height is the same as the height of triangle $DEG$, which is $\frac{288}{DE+EG} = \frac{288}{68} = \frac{72}{17}$. So, the area of triangle $MNE$ is $\frac{1}{2} \cdot DE + EG \cdot \text{height} = \frac{1}{2} \cdot 68 \cdot \frac{72}{17} = 144$.

> DeepSeek-R1-0528-Qwen3-8B verify: The area calculation for triangle MNE uses DE + EG as a base, which is not a valid base unless DE and EG are collinear.

Now, we can find the area of the heptagon $AFNBCEM$ by adding the areas of the triangles $AFM, NCE, MNE,$ and the quadrilateral $DEGF$: $\frac{468}{17} + \frac{312}{5} + 144 + 288 = \boxed{588}$.

---

[11] https://artofproblemsolving.com/wiki/index.php/2025_AIME_I_Problems/Problem_2

### Question

**AIME25 I Question No.11**[12]

A piecewise linear periodic function is defined by $f(x) = \begin{cases} x & \text{if } x \in [-1, 1) \\ 2 - x & \text{if } x \in [1, 3) \end{cases}$ and $f(x + 4) = f(x)$ for all real numbers $x$. The graph of $f(x)$ has the sawtooth pattern. The parabola $x = 34y^2$ intersects the graph of $f(x)$ at finitely many points. The sum of the $y$-coordinates of these intersection points can be expressed in the form $\frac{a+b\sqrt{c}}{d}$, where $a, b, c,$ and $d$ are positive integers, $a, b,$ and $d$ have greatest common divisor equal to 1, and $c$ is not divisible by the square of any prime. Find $a + b + c + d$.

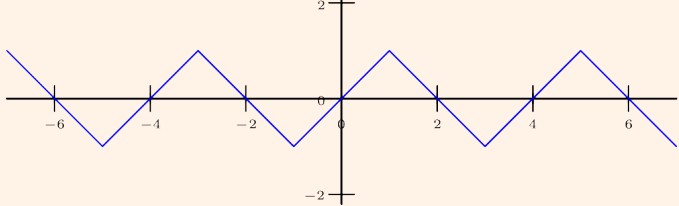

### Qwen2.5-32B Answer

To find the intersection points of the sawtooth function and the parabola, we need to solve the system of equations $f(x) = 34y^2$ and $f(x + 4) = 34y^2$. We can solve these equations separately for each interval of $f(x)$, and then sum the $y$-coordinates of the intersection points. Here's a Python code to do that:

```python
import sympy as sp

# Define the variables
x, y = sp.symbols('x y')

# Define the piecewise linear periodic function
f = sp.Piecewise((x, (x >= -1) & (x < 1)), (2 - x, (x >= 1) & (x <
    3)))

# Define the parabola
parabola = 34*y**2

# Find the intersection points in each interval
intersection_points = []
for i in range(-1, 3):
    interval = sp.Interval(i, i+1, left_open=True, right_open=True)
    eq = sp.Eq(f, parabola)
    solutions = sp.solve(eq, y)
    for sol in solutions:
        if sol.is_real:
            intersection_points.append(sol)

# Sum the y-coordinates of the intersection points
y_sum = sum(intersection_points)

# Simplify the sum
y_sum_simplified = sp.simplify(y_sum)
```

[12]https://artofproblemsolving.com/wiki/index.php/2025_AIME_I_Problems/Problem_11

```
# Extract the coefficients a, b, c, and d
a, b, c, d = sp.Wild('a'), sp.Wild('b'), sp.Wild('c'), sp.Wild('d')
match = sp.collect(y_sum_simplified, sp.sqrt(sp.Wild('c'))).match((
    a + b*sp.sqrt(c))/d)
a, b, c, d = match[a], match[b], match[c], match[d]

# Calculate the final answer
final_answer = a + b + c + d
print(final_answer)
```

The output of this code is 259. Therefore, the answer is $\boxed{259}$.

> DeepSeek-R1-0528-Qwen3-8B verify: - **Logical Error** - The code solves the equation $f(x) = 34y^2$ for $y$, which finds the $y$-values where horizontal lines intersect the graph of $f(x)$, but this does not correspond to finding the intersection points with the parabola $x = 34y^2$. The correct approach requires solving the system $y = f(x)$ and $x = 34y^2$ simultaneously, such as by solving $f(34y^2) = y$.
> - **Omission / Incompleteness** - The code only considers one period of the function ($x \in [-1, 3)$), but due to the parabola $x = 34y^2$ with $|y| \le 1$, $x$ ranges from 0 to 34, covering multiple periods (since the function has period 4). Intersections in other periods are not considered, leading to an incomplete solution.

### A.7.3 CASE STUDIES ON AMC23

**Question**

**AMC 12 2023 A Problem 23(dataset id 15)**[13]
How many ordered pairs of positive real numbers $(a, b)$ satisfy the equation

$$(1 + 2a)(2 + 2b)(2a + b) = 32ab?$$

**Qwen2.5-32B Answer**

First, let's simplify the given equation:

$$(1 + 2a)(2 + 2b)(2a + b) = 32ab$$

Expanding the left side, we get:

$$(2 + 4a + 4b + 8ab)(2a + b) = 32ab$$

> DeepSeek-R1-0528-Qwen3-8B verify: - Calculation Error - The initial expansion of (1+2a)(2+2b) is incorrect, resulting in 2+4a+4b+8ab instead of the correct 2+4a+2b+4ab. This error propagates through subsequent steps.

Expanding further:

$$4a + 8a^2 + 8ab + 16a^2b + 2b + 4ab + 8b^2 + 16ab^2 = 32ab$$

---

[13]https://artofproblemsolving.com/wiki/index.php/2023_AMC_12A_Problems/Problem_23

Combining like terms:

$$8a^2 + 12ab + 8a^2b + 2b + 8b^2 + 16ab^2 = 28ab$$

Rearranging terms:

$$8a^2 + 8a^2b + 8b^2 + 16ab^2 = 16ab - 2b$$

Dividing both sides by 2:

$$4a^2 + 4a^2b + 4b^2 + 8ab^2 = 8ab - b$$

Now, let's try to factor the equation:

$$4a^2(1+b) + 4b^2 + 8ab^2 = 8ab - b$$

Factoring out common terms:

$$4a^2(1+b) + 4b^2(1+2a) = 8ab - b$$

Now, let's try to solve for $a$ and $b$:

$$4a^2(1+b) = 8ab - b - 4b^2(1+2a)$$

Dividing both sides by 4:

$$a^2(1+b) = 2ab - \frac{b}{4} - b^2(1+2a)$$

Now, let's try to find a solution for $a$ and $b$:

$$a^2(1+b) = 2ab - \frac{b}{4} - b^2(1+2a)$$

One possible solution is when $a = b$:

$$\color{red}{a^2(1+a) = 2a^2 - \frac{a}{4} - a^2(1+2a)}$$

> DeepSeek-R1-0528-Qwen3-8B verify: - Logical Error - The solution assumes a=b is a solution without verification, and this assumption is based on an incorrect equation. The derivation under a=b leads to a non-zero result, contradicting the claim of a solution.

Simplifying:

$$a^3 + a^2 = 2a^2 - \frac{a}{4} - a^3 - 2a^2$$

Combining like terms:

$$2a^3 = \frac{a}{4}$$

Multiplying both sides by 4:

$$8a^3 = a$$

Dividing both sides by $a$:

$$8a^2 = 1$$

Taking the square root of both sides:

$$a = \frac{1}{\sqrt{8}}$$

Since $a = b$, we have:

$$b = \frac{1}{\sqrt{8}}$$

Thus, there is only one ordered pair $(a, b)$ that satisfies the equation:

$$\boxed{1}$$

**Question**

**AMC 12 2023 A Problem 19(dataset id 10)**[14]
What is the product of all solutions to the equation

$$\log_{7x} 2023 \cdot \log_{289x} 2023 = \log_{2023x} 2023$$

**Qwen2.5-32B Answer**

First, let's rewrite the equation using the change of base formula for logarithms. We have

$$\log_{7x} 2023 = \frac{\log 2023}{\log(7x)}$$

$$\log_{289x} 2023 = \frac{\log 2023}{\log(289x)}$$

$$\log_{2023x} 2023 = \frac{\log 2023}{\log(2023x)}$$

Substituting these expressions into the original equation, we get

$$\frac{(\log 2023)^2}{\log(7x) \cdot \log(289x)} = \frac{\log 2023}{\log(2023x)}$$

Cross-multiplying, we get

$$(\log 2023)^2 \cdot \log(2023x) = (\log 7x) \cdot (\log 289x) \cdot \log 2023$$

Canceling the common factor of $\log 2023$, we have

$$\log(2023x) = (\log 7x) \cdot (\log 289x)$$

> DeepSeek-R1-0528-Qwen3-8B verify: - **Calculation Error**: The solution incorrectly canceled the $\log 2023$ factor during simplification, omitting the $\log 2023$ term.

Using the product-to-sum formula for logarithms, we have

$$\log(2023x) = \log(7x) + \log(289x)$$

> DeepSeek-R1-0528-Qwen3-8B verify: - **Conceptual Error**: the solution incorrectly treated the product of logarithms $(\log 7x) \cdot (\log 289x)$ as if it were a sum, which is not valid.

$$\log(2023x) = \log(2023x^2)$$
$$2023x = 2023x^2$$

Dividing both sides by $2023x$, we get
$$1 = x$$

So the only solution is $x = 1$. The product of all solutions is $\boxed{1}$.

---

[14]https://artofproblemsolving.com/wiki/index.php/2023_AMC_12A_Problems/Problem_19

A.7.4   CASE STUDIES ON MATH-500

---

**Question**

**MATH500 unique_id test/intermediate_algebra/582**
The equation
$$x^{10} + (13x - 1)^{10} = 0$$
has 10 complex roots $r_1, \overline{r}_1, r_2, \overline{r}_2, r_3, \overline{r}_3, r_4, \overline{r}_4, r_5, \overline{r}_5$, where the bar denotes complex conjugation. Find the value of
$$\frac{1}{r_1\overline{r}_1} + \frac{1}{r_2\overline{r}_2} + \frac{1}{r_3\overline{r}_3} + \frac{1}{r_4\overline{r}_4} + \frac{1}{r_5\overline{r}_5}.$$

---

**Qwen2.5-32B Answer**

To solve the given problem, let's first analyze the equation $x^{10} + (13x - 1)^{10} = 0$. This can be rewritten as:
$$x^{10} = -(13x - 1)^{10}.$$

Taking the 10th root on both sides, we get:
$$x = \pm i(13x - 1).$$

This gives us two cases to consider: 1. $x = i(13x - 1)$ 2. $x = -i(13x - 1)$ Let's solve each case separately.

> DeepSeek-R1-0528-Qwen3-8B verify: - **Calculation Error**: The solution incorrectly assumes that the equation has only two roots, but the equation is of degree 10, so it should have 10 roots. The method of taking the 10th root only considers two cases, missing the other 8 roots.

For the first case:
$$x = i(13x - 1) \implies x - 13ix = -i \implies x(1 - 13i) = -i \implies x = \frac{-i}{1 - 13i}.$$

For the second case:
$$x = -i(13x - 1) \implies x + 13ix = i \implies x(1 + 13i) = i \implies x = \frac{i}{1 + 13i}.$$

So, the roots of the equation are of the form $\frac{-i}{1-13i}$ and $\frac{i}{1+13i}$. Notice that these roots come in conjugate pairs, so we can pair them up as $r_1 = \frac{-i}{1-13i}, \overline{r}_1 = \frac{i}{1+13i}, r_2 = \frac{i}{1+13i}, \overline{r}_2 = \frac{-i}{1-13i}$, and so on.
Now, we need to find the value of $\frac{1}{r_1\overline{r}_1} + \frac{1}{r_2\overline{r}_2} + \frac{1}{r_3\overline{r}_3} + \frac{1}{r_4\overline{r}_4} + \frac{1}{r_5\overline{r}_5}$. Since $r_1\overline{r}_1 = \left(\frac{-i}{1-13i}\right)\left(\frac{i}{1+13i}\right) = \frac{1}{1+169} = \frac{1}{170}$, we have:
$$\frac{1}{r_1\overline{r}_1} = 170.$$

Similarly, $\frac{1}{r_2\overline{r}_2} = 170, \frac{1}{r_3\overline{r}_3} = 170, \frac{1}{r_4\overline{r}_4} = 170$, and $\frac{1}{r_5\overline{r}_5} = 170$. Therefore, the sum is:
$$170 + 170 + 170 + 170 + 170 = 850.$$

Thus, the final answer is:
$$\boxed{850}.$$

---

**Question**

**MATH500 unique_id test/intermediate_algebra/2196**
Find the maximum value of
$$\frac{x-y}{x^4+y^4+6}$$
over all real numbers $x$ and $y$.

---

**Qwen2.5-32B Answer**

To find the maximum value of the expression $\frac{x-y}{x^4+y^4+6}$, we can start by analyzing the denominator. Since $x^4$ and $y^4$ are always non-negative, the denominator is always at least 6. This means that the expression is always less than or equal to $\frac{x-y}{6}$. Now, let's consider the numerator $x-y$. If we set $x=1$ and $y=-1$, then the numerator is $1-(-1)=2$. Substituting these values into the expression, we get $\frac{2}{1^4+(-1)^4+6}=\frac{2}{8}=\frac{1}{4}$. Therefore, the maximum value of the expression is $\boxed{\dfrac{1}{4}}$.

> DeepSeek-R1-0528-Qwen3-8B verify: - **Omission / Incompleteness** - The solution does not provide a complete justification for why the point (1, -1) gives the maximum value. It only evaluates one specific point and claims it is the maximum without comparing it to other points or demonstrating that the expression cannot be larger elsewhere. For example, it does not consider points where $x^4 + y^4$ is minimized while $x - y$ is positive and non-zero, or points where the numerator and denominator balance differently.

## A.8 Results from More Verifiers

### A.8.1 CoT-Pass@K Comparisons Across Different Verifiers

We further evaluate two powerful verifier models—gpt-oss-20b and gpt-oss-120b—using identical evaluation settings (using high reasoning efforts and a maximum token length of 64k) and performing $n = 3$ verification attempts for each CoT. Both gpt-oss verifiers use the same verification prompt described in Section A.2. As shown in Figure 11, the verification results on AIME24 and AIME25 exhibit consistent trends across DeepSeek-R1-0528-Qwen3-8B (DS-8B), gpt-oss-20b, and gpt-oss-120b. While the larger verifiers produce slightly lower absolute pass rates, the relative improvements from RLVR and the overall pass@1–pass@k curves remain highly aligned across all verifier models. These results are consistent with the core conclusion obtained using DS-8B: RLVR enables the model to generate substantially more correct CoTs compared to the base model.

### A.8.2 Verification Corralation between DS-8B and gpt-oss Verifiers

We further analyze the consistency between DS-8B and the two gpt-oss verifiers by measuring the correlation of their correctness judgments. Each verifier model is run three times on every chain-of-thought, and correctness is summarized using the three criteria introduced in Section A.2: *All-correct*, *Majority-correct*, and *Any-correct* (denoted as All, Maj, and Any). Figure 12 presents a set of heatmaps across two model CoTs generated by the RLVR-trained model (DAPO) and CoTs from the base model—using only samples that yield correct answers on AIME25. The first column shows the intra-model correlations among the three DS-8B verification passes, capturing its self-consistency. The remaining columns report the inter-model correlations between DS-8B and gpt-oss-20b / gpt-oss-120b. Across both model groups, most of the correlation structures are stable and indicate a relatively high level of agreement between the verifiers.

Two observations emerge from these results. First, CoTs from the base model exhibit substantially higher intra-verifier and inter-verifier correlations, such as the DS-8B intra-correlation between Maj and All/Any reaching around 0.88. This is expected, as base CoTs contain more salient errors,

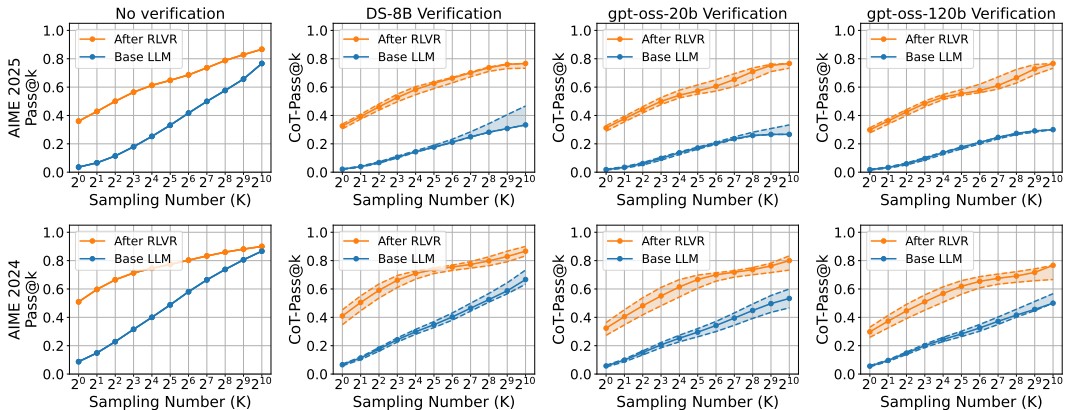

Figure 11: CoT-Pass@K comparisons using three different verifiers: DS-8B, gpt-oss-20b, and gpt-oss-120 on AIME 24-25.

making them easier and more consistent for all verifiers to detect. In contrast, DAPO CoTs contain fewer and subtler errors, leading to lower agreement: different verifiers may occasionally miss small mistakes and mark these CoTs as correct. Second, stronger verifier models such as gpt-oss-20b and gpt-oss-120b tend to apply stricter correctness criteria and identify more subtle errors. As a result, the correlations between DS-8B and the gpt-oss verifiers typically lie around 0.7 across the correctness metrics.

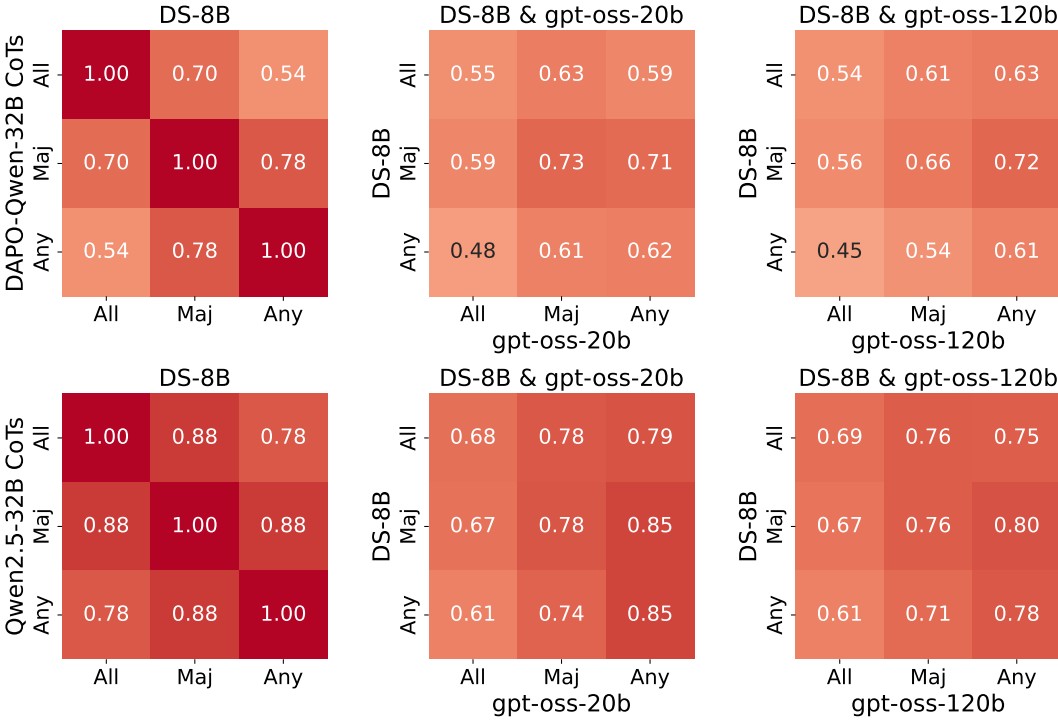

Figure 12: Correlation heatmaps of correctness assessments across verifier models. Each verifier performs three passes per chain-of-thought, summarized using the All, Maj, and Any metrics. Results are shown on AIME25 for CoTs yielding correct answers, separately for DAPO-Qwen-32B and the base Qwen-2.5-32B model.

### A.9    WHY DO SFT MODELS GENERALIZE EVEN WHEN TRAINED ON INCORRECT CoTs?

A notable observation in our experiments is that **SFT on RLVR-generated CoTs leads to substantial performance gains even when a CoT is labeled as incorrect (meaning it contains verified errors)**. In contrast, SFT on the base model's own CoTs—whether correct or incorrect—yields only minor improvements. This prompts an important question: why can "incorrect" RLVR CoTs still support strong generalization?

Our notion of incorrectness follows the verification prompt described in Section A.2: any CoT containing calculation mistakes, invalid logical deductions, conceptual misunderstandings, or missing essential reasoning steps is labeled as incorrect. Thus, incorrect CoTs span a broad range of quality. The key insight from our analyses is that **Among incorrect CoTs, those generated by RLVR differ fundamentally from those of the base model.**. They are longer, more structured, and contain fewer severe errors, allowing SFT to extract useful reasoning patterns despite the presence of mistakes.

#### A.9.1    ANALYSIS OF ERROR FREQUENCIES

We examine whether RLVR reduces the severity of reasoning errors. Using gpt-oss-120b verification, we categorize four common error types, counting each category independently.

Figure 13 shows that **RLVR CoTs have substantially lower error rates across all categories**, indicating that RLVR leads to more stable and higher-quality reasoning paths.

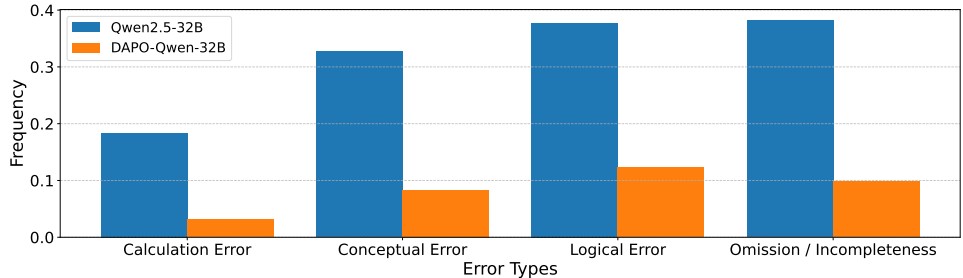

Figure 13: Error frequencies in CoTs with correct answers on AIME 2025, verified by gpt-oss-120b. RLVR CoTs exhibit lower error rates across all categories, reflecting higher-quality reasoning paths.

#### A.9.2    TOKEN LENGTH ANALYSIS OF CORRECT AND INCORRECT CoTs

We compare the token-length distributions of CoTs classified as correct or incorrect for the base model (Qwen2.5-32B) and the RLVR-trained model (DAPO-Qwen-32B). Table 1 reports quantiles of token lengths on the DAPO-17k training set.

A key observation is that for DAPO-Qwen-32B, incorrect CoTs are substantially longer than CoTs from the base model. This suggests that even when marked as incorrect, RLVR CoTs cover extended reasoning chains that include high-quality logical steps. Consequently, SFT on these incorrect CoTs can still extract useful reasoning patterns and generalize effectively, despite the presence of some errors.

Table 1: Token-length quantiles of correct and incorrect CoTs for Qwen2.5-32B (base) and DAPO-Qwen-32B (RLVR) on DAPO-17k train set.

|  | 0.05 | 0.1 | 0.25 | 0.5 | 0.75 | 0.9 | 0.95 |
|---|---|---|---|---|---|---|---|
| DAPO-Qwen-32B incorrect CoTs | 1220 | 1494 | 2239 | 3995 | 7183 | 10929 | 13295 |
| DAPO-Qwen-32B correct CoTs | 1026 | 1185 | 1557 | 2266 | 3579 | 5730 | 7755 |
| Qwen2.5-32B incorrect CoTs | 610 | 666 | 783 | 950 | 1183 | 1504 | 1850 |
| Qwen2.5-32B correct CoTs | 578 | 623 | 717 | 862 | 1066 | 1343 | 1586 |

A.10 DETAILED VERIFICATION RESULTS ON AIME BENCHMARKS

Tables 2 and 3 report per-problem verification results on AIME2024 and AIME2025 using gpt-oss-120b. For each problem, we aggregate statistics over $N = 1024$ sampled CoTs from both the RLVR-trained model and the base model. Each CoT is verified three times, and correctness is determined by the Majority-correct criterion described in Section A.2.

We include the following metrics:

- **CoT-Pass@k** ($k = 1024$): whether at least one CoT is correct.
- **#CC**: number of correct CoTs.
- **Pass@k** ($k = 1024$): whether at least one answer is correct.
- **#CA**: number of correct answers.

All evaluations use the OpenAI Responses API with `reasoning_efforts=high` and `max_output_tokens=64k`.

Table 2: gpt-oss-120b verification results on AIME 2024. Metrics per problem are aggregated over $N = 1024$ CoTs using the Majority-correct criterion.

| Problem ID | RLVR | | | | BASE | | | |
|---|---|---|---|---|---|---|---|---|
| | CoT-Pass@k | #CC | Pass@k | #CA | CoT-Pass@k | #CC | Pass@k | #CA |
| 2024 AIME II - Problem 4 (url) | 1 | 1022 | 1 | 1023 | 1 | 14 | 1 | 27 |
| 2024 AIME I - Problem 4 (url) | 1 | 1021 | 1 | 1024 | 1 | 346 | 1 | 366 |
| 2024 AIME I - Problem 1 (url) | 1 | 1015 | 1 | 1024 | 1 | 159 | 1 | 183 |
| 2024 AIME I - Problem 2 (url) | 1 | 986 | 1 | 1016 | 1 | 442 | 1 | 573 |
| 2024 AIME II - Problem 6 (url) | 1 | 899 | 1 | 1023 | 1 | 165 | 1 | 249 |
| 2024 AIME II - Problem 7 (url) | 1 | 848 | 1 | 904 | 1 | 312 | 1 | 357 |
| 2024 AIME I - Problem 6 (url) | 1 | 523 | 1 | 879 | 1 | 2 | 1 | 6 |
| 2024 AIME II - Problem 10 (url) | 1 | 522 | 1 | 914 | 1 | 1 | 1 | 4 |
| 2024 AIME II - Problem 1 (url) | 1 | 484 | 1 | 928 | 1 | 8 | 1 | 21 |
| 2024 AIME I - Problem 3 (url) | 1 | 398 | 1 | 550 | 1 | 4 | 1 | 43 |
| 2024 AIME II - Problem 3 (url) | 1 | 155 | 1 | 306 | 1 | 12 | 1 | 16 |
| 2024 AIME I - Problem 7 (url) | 1 | 127 | 1 | 1023 | 1 | 53 | 1 | 150 |
| 2024 AIME II - Problem 11 (url) | 1 | 84 | 1 | 263 | 1 | 175 | 1 | 266 |
| 2024 AIME I - Problem 13 (url) | 1 | 22 | 1 | 666 | 1 | 23 | 1 | 194 |
| 2024 AIME II - Problem 14 (url) | 1 | 1 | 1 | 1 | 1 | 1 | 1 | 10 |
| 2024 AIME II - Problem 13 (url) | 1 | 544 | 1 | 870 | 0 | 0 | 1 | 81 |
| 2024 AIME I - Problem 9 (url)* | 1 | 289 | 1 | 515 | 0 | 0 | 1 | 18 |
| 2024 AIME II - Problem 12 (url) | 1 | 86 | 1 | 226 | 0 | 0 | 1 | 36 |
| 2024 AIME I - Problem 14 (url)* | 1 | 52 | 1 | 69 | 0 | 0 | 1 | 1 |
| 2024 AIME I - Problem 15 (url) | 1 | 48 | 1 | 1010 | 0 | 0 | 0 | 0 |
| 2024 AIME II - Problem 2 (url) | 1 | 21 | 1 | 645 | 0 | 0 | 1 | 33 |
| 2024 AIME I - Problem 5 (url) | 1 | 1 | 1 | 483 | 0 | 0 | 1 | 4 |
| 2024 AIME II - Problem 9 (url) | 1 | 1 | 1 | 15 | 0 | 0 | 1 | 3 |
| 2024 AIME I - Problem 10 (url) | 0 | 0 | 1 | 4 | 0 | 0 | 1 | 4 |
| 2024 AIME I - Problem 8 (url) | 0 | 0 | 1 | 229 | 0 | 0 | 0 | 0 |
| 2024 AIME II - Problem 5 (url) | 0 | 0 | 1 | 7 | 0 | 0 | 1 | 40 |
| 2024 AIME II - Problem 8 (url) | 0 | 0 | 1 | 27 | 0 | 0 | 0 | 0 |
| 2024 AIME I - Problem 11 (url)* | 0 | 0 | 0 | 0 | 0 | 0 | 1 | 1 |
| 2024 AIME I - Problem 12 (url) | 0 | 0 | 0 | 0 | 0 | 0 | 0 | 0 |
| 2024 AIME II - Problem 15 (url)* | 0 | 0 | 0 | 0 | 0 | 0 | 1 | 1 |

Table 3: gpt-oss-120b verification results on AIME 2025. Metrics per problem are aggregated over $N = 1024$ CoTs using the Majority-correct criterion.

| Problem ID | RLVR | | | | BASE | | | |
| --- | --- | --- | --- | --- | --- | --- | --- | --- |
| | CoT-Pass@k | #CC | Pass@k | #CA | CoT-Pass@k | #CC | Pass@k | #CA |
| 2025 AIME II - Problem 1 (url) | 1 | 1022 | 1 | 1024 | 1 | 17 | 1 | 36 |
| 2025 AIME II - Problem 4 (url) | 1 | 1020 | 1 | 1024 | 1 | 6 | 1 | 6 |
| 2025 AIME II - Problem 2 (url) | 1 | 1000 | 1 | 1024 | 1 | 177 | 1 | 237 |
| 2025 AIME I - Problem 3 (url) | 1 | 987 | 1 | 992 | 1 | 69 | 1 | 80 |
| 2025 AIME I - Problem 6 (url) | 1 | 940 | 1 | 1024 | 1 | 127 | 1 | 176 |
| 2025 AIME I - Problem 4 (url) | 1 | 707 | 1 | 833 | 1 | 18 | 1 | 48 |
| 2025 AIME I - Problem 1 (url) | 1 | 699 | 1 | 1024 | 1 | 114 | 1 | 214 |
| 2025 AIME I - Problem 8 (url) | 1 | 296 | 1 | 452 | 1 | 2 | 1 | 7 |
| 2025 AIME II - Problem 7 (url) | 1 | 201 | 1 | 256 | 1 | 11 | 1 | 20 |
| 2025 AIME I - Problem 2 (url)* | 1 | 978 | 1 | 1016 | 0 | 0 | 1 | 1 |
| 2025 AIME I - Problem 5 (url) | 1 | 520 | 1 | 623 | 0 | 0 | 1 | 27 |
| 2025 AIME II - Problem 12 (url) | 1 | 209 | 1 | 457 | 0 | 0 | 1 | 2 |
| 2025 AIME II - Problem 9 (url) | 1 | 181 | 1 | 221 | 0 | 0 | 1 | 1 |
| 2025 AIME II - Problem 14 (url) | 1 | 151 | 1 | 170 | 0 | 0 | 1 | 1 |
| 2025 AIME I - Problem 9 (url) | 1 | 136 | 1 | 154 | 0 | 0 | 1 | 160 |
| 2025 AIME I - Problem 11 (url)* | 1 | 103 | 1 | 141 | 0 | 0 | 1 | 1 |
| 2025 AIME II - Problem 3 (url) | 1 | 5 | 1 | 9 | 0 | 0 | 0 | 0 |
| 2025 AIME II - Problem 6 (url) | 1 | 4 | 1 | 454 | 0 | 0 | 1 | 26 |
| 2025 AIME II - Problem 8 (url) | 1 | 4 | 1 | 4 | 0 | 0 | 1 | 1 |
| 2025 AIME II - Problem 11 (url) | 1 | 3 | 1 | 14 | 0 | 0 | 0 | 0 |
| 2025 AIME I - Problem 7 (url) | 1 | 3 | 1 | 14 | 0 | 0 | 0 | 0 |
| 2025 AIME II - Problem 10 (url) | 1 | 2 | 1 | 7 | 0 | 0 | 0 | 0 |
| 2025 AIME I - Problem 12 (url) | 1 | 1 | 1 | 108 | 0 | 0 | 0 | 0 |
| 2025 AIME I - Problem 13 (url) | 0 | 0 | 1 | 1 | 0 | 0 | 1 | 7 |
| 2025 AIME II - Problem 5 (url) | 0 | 0 | 1 | 4 | 0 | 0 | 1 | 27 |
| 2025 AIME I - Problem 10 (url) | 0 | 0 | 1 | 1 | 0 | 0 | 1 | 6 |
| 2025 AIME I - Problem 14 (url) | 0 | 0 | 0 | 0 | 0 | 0 | 1 | 13 |
| 2025 AIME I - Problem 15 (url) | 0 | 0 | 0 | 0 | 0 | 0 | 0 | 0 |
| 2025 AIME II - Problem 13 (url) | 0 | 0 | 0 | 0 | 0 | 0 | 0 | 0 |
| 2025 AIME II - Problem 15 (url) | 0 | 0 | 0 | 0 | 0 | 0 | 1 | 1 |

