# OpenReview forum: "Reinforcement Learning with Verifiable Rewards Implicitly Incentivizes Correct Reasoning in Base LLMs"
_ICLR.cc/2026/Conference — ICLR 2026 Poster_

### Official Review · Reviewer_jCat · 2025-10-17

**Soundness:** 2
**Presentation:** 3
**Contribution:** 3
**Rating:** 6
**Confidence:** 4

**Summary:**

This manuscript investigates a fundamental open question in reinforcement learning with verifiable rewards (RLVR): does RLVR merely improve sampling efficiency, or does it endow the base model with fundamentally new reasoning capabilities? The authors first revisit the Pass@K metric, which the RL community has often considered a signal of exploration, and identify its potential pitfalls. Specifically, base LLMs are capable of producing incorrect Chains of Thought (CoTs) yet coincidentally arriving at the ground truth. To address this, they introduce a new metric, CoT-Pass@K, which simultaneously considers the correctness of both the reasoning trace and the final answer. By re-examining the model's reasoning performance with CoT-Pass@K, the authors observe that RLVR successfully extends the reasoning boundary of the base model in both code and math tasks. The manuscript also presents primary theorems on GRPO's advantage estimation to interpret these empirical observations.

**Strengths:**

- The exact role of RLVR in enhancing the reasoning capabilities of base models is an open and fundamental problem for the RL community. Thus, this manuscript begins with a very strong motivation and is, in my view, highly timely.

- Although the proposed Theorem 1 is relatively simple and straightforward, this preliminary explanation is helpful for the community to understand the underlying reasons why RLVR works.

- The authors base their conclusions on extensive experiments. I particularly appreciate that they not only experimented with the Qwen+Math setup but also conducted additional experiments on coding tasks.

**Weaknesses:**

- The correctness of a CoT is determined by a R1-distilled-Qwen3-8B model, which may not be a sufficiently strong verifier. Since reasoning models may frequently exhibit rethinking or backtracking behavior, I have concerns about whether an 8B model is reliable enough to verify complex reasoning traces that include self-correction, reflection, and backtracking. For example, what if a model initially thinks incorrectly but then corrects its answer on its own? Is the distilled 8B model strong enough to verify such a CoT? The authors acknowledge this limitation in line 468, but I still feel it is noteworthy. Stronger frontier close-source LLMs maybe a better choice, e.g., Gemini-2.5-pro, if the cost of API calls is affordable.

- The empirical observations from the post-SFT experiments (lines 429-431) are difficult for me to follow. It is counterintuitive to me that: 1) tuning on erroneous CoTs can also improve Pass@1 accuracy, ultimately leading to nearly the same performance, and 2) SFT on all CoTs results in worse accuracy than SFT on only incorrect CoTs. Does this observation indicate that the "incorrect" CoTs identified by Distilled-Qwen3-8B only contain erroneous steps at the beginning (which the model self-corrects later) rather than the entire reasoning trace being wrong? And thus SFT on these partially incorrect reasoning traces can also benefits pass@1 accuracy. If this is true, I have further concerns about the reliability of the CoT verifier and the definition of a "correct" CoT.

In summary, I am positive about this manuscript, and I find its findings inspiring and important. However, due to the two concerns mentioned above, I do not have enough confidence to give it a higher rating.

**Questions:**

1. How would the authors define the correctness of a CoT? That is, if a model makes mistakes initially but then corrects them in subsequent tokens through self-reflection, should such a CoT be considered incorrect?

2. It would be good to see the influence of SFT on both Pass@K and CoT-Pass@K. For example, DAPO-Qwen-32B and DeepSeek-R1-Distilled-Qwen-32B have similar avg@32 on AIME24. It would be interesting to see which model performs better when using CoT-Pass@K as a metric.

---

> ### Author Response · Authors · 2025-11-20
> **Thank you for your feedback**
>
> ### **Response to Weakness 1 and Question 1: Reliable Estimation of CoT-Pass@K**
>
> > The correctness of a CoT is determined by a R1-distilled-Qwen3-8B model, which may not be a sufficiently strong verifier. Since reasoning models may frequently exhibit rethinking or backtracking behavior, I have concerns about whether an 8B model is reliable enough to verify complex reasoning traces that include self-correction, reflection, and backtracking. For example, what if a model initially thinks incorrectly but then corrects its answer on its own? Is the distilled 8B model strong enough to verify such a CoT? The authors acknowledge this limitation in line 468, but I still feel it is noteworthy. Stronger frontier close-source LLMs maybe a better choice, e.g., Gemini-2.5-pro, if the cost of API calls is affordable.
>
> > How would the authors define the correctness of a CoT? That is, if a model makes mistakes initially but then corrects them in subsequent tokens through self-reflection, should such a CoT be considered incorrect?
>
> We deeply appreciate your detailed scrutiny regarding the reliability of our verification process. We understand your concern: if the verifier (8B) is smaller than the generator (32B), can it reliably distinguish between "hallucinated reasoning" and complex, valid behaviors like self-correction?
>
> To address this, we clarify our definition of correctness and provide three layers of evidence: validation via stronger models, extensive manual auditing, and inference capacity justification.
>
> 1. Definition of CoT Correctness & Handling Self-Correction
>
> We define a "correct" CoT as a reasoning trace where the intermediate steps contain accurate and complete logic sufficient to derive the final answer. Tt is rather challenging to automatically judge whether a CoT satisfies this condition. In our specific implementation (see Appendix A.2), we instruct the verifier to identify four specific types of fatal flaws:
>
> - **Calculation Error:** Mistakes in arithmetic or algebraic manipulation.
> - **Logical Error:** Invalid inferences or flawed deductions.
> - **Conceptual Error:** Misunderstanding of mathematical definitions.
> - **Omission:** Missing steps or incomplete justification.
>
> Then if there is no error identified by the verifier, we treat the CoT as correct. In practice, we observe this mechanism is good at capturing intermidate errors precisely but may miss some, meaning the error finding is typically high precision yet relatively low recall. Thus we verify each CoT at least three times and employ any-correct, all-correct, majority-correct strategies to provide robust verification results.
>
> Regarding **self-correction and backtracking**: Our prompt instructs the verifier to evaluate the derivation's validity as a whole. If a model makes an initial error but successfully catches and corrects it within the trace (resulting in a logically sound path), a capable verifier should not flag this as a "Logical Error." However, we acknowledge that distinguishing between *valid self-correction* and *incoherent rambling* is difficult. The verifier may make also mistakes. We have conducted some manual checks on this issue and did not find it affects the reliable verification. What we find relatively unstable is that the verifier may miss some reasoning errors across different queries, once it confirms a reasoning error, our manual check typically agrees there is a mistake.
>
> 2. Validation via Stronger Verifiers (New Experiments)
>
> We followed your advice to validate our findings using more capable models available and employed gpt-oss-20b and gpt-oss-120b (operating in high-reasoning effort mode) as verifiers.
> Specifically, gpt-oss-120b can be almost regarded as a perfect solver on AIME 24-25.
>
> - **CoT-Pass@K in Different Verifiers:** As shown in **Figure 11 (Appendix A.8)**, the trends observed with these stronger verifiers corroborate our main claims. While `gpt-oss-120b` is strictly more rigorous—identifying subtle reasoning errors that the 8B model might miss—the relative performance gaps and the behavior of RLVR models remain consistent. This confirms that our results are not an artifact of a weak verifier.
> - **Comparisons Across Verifiers:** These LLM verifiers typically have high error-detection precision yet relatively low recall. Therefore, using multiple verifications to aggregate results is a necessary step. All these verifiers can effectively identify fatal errors in base LLMs, while may not be that consistent when analyzing the CoTs from post-RLVR models, such as DAPO-32B. We find gpt-oss-120b to be the most strict and rigorous in detecting reasoning errors. Moreover,

---

> > ### Author Response · Authors · 2025-11-20
> >
> > 3. Extensive Manual Verification (No Cherry-Picking)
> >
> > To address the concern that we might be misinterpreting "self-correction" as "guessing," we conducted a rigorous manual audit.
> >
> > - **Tables 2 and 3 (Appendix):** We provide a comprehensive summary of `Pass@K` and `CoT-Pass@K` ($K=1024$) for *every* question in AIME 24-25.
> > - **Appendix A.7:** We provide detailed case studies of questions that exhibit **High Pass@K but Low CoT-Pass@K**. These are *not* instances of valid self-correction; they are clear instances where the base model "hacks" or guesses the answer using logically bankrupt derivation.
> > - **Open Data Promise:** To ensure transparency, we will publicly release all reasoning traces and verification results for AIME 24-25, allowing the community to perform the same per-question analysis.
> >
> > 4. Inference Capacity Justification: Test-Time Scaling
> >
> > Finally, we offer a inference-capacity perspective on why the 8B verifier (DeepSeek-R1-Distill-Qwen-8B) is effective against a 32B base model.
> >
> > - **Effective Inference Capacity:** In reasoning tasks, effective capacity is a function of $Parameters \times GeneratedTokens$.
> > - **Comparison:** The 8B verifier utilizes a long-context reasoning mode (often consuming ~$8B \times 64K$ tokens of compute), whereas `Qwen-32B-Base` is typically bounded by standard generation ($32B \times 4K$ tokens).
> > - **Result:** Through this "test-time scaling," the distilled 8B model achieves a reasoning depth sufficient to decompose and audit the shorter, shallower traces of the 32B base model. While verifying `DAPO-32B` is more challenging, the verifier remains informative because it is specifically distilled on solving math problems and is specialized at detecting the kinds of inherent logical errors that persist in the 32B model.

---

> ### Author Response · Authors · 2025-11-20
>
> ### Response to Weakness 2: Post-SFT Observations and Verifier Reliability
>
> > The empirical observations from the post-SFT experiments (lines 429-431) are difficult for me to follow. It is counterintuitive to me that: 1) tuning on erroneous CoTs can also improve Pass@1 accuracy, ultimately leading to nearly the same performance, and 2) SFT on all CoTs results in worse accuracy than SFT on only incorrect CoTs. Does this observation indicate that the "incorrect" CoTs identified by Distilled-Qwen3-8B only contain erroneous steps at the beginning (which the model self-corrects later) rather than the entire reasoning trace being wrong? And thus SFT on these partially incorrect reasoning traces can also benefits pass@1 accuracy. If this is true, I have further concerns about the reliability of the CoT verifier and the definition of a "correct" CoT.
>
>
> We appreciate your sharp observation regarding the counterintuitive results in the post-SFT experiments. We agree that, at first glance, it seems contradictory that fine-tuning on "incorrect" CoTs could yield performance gains comparable to tuning on "correct" ones. However, a granular qualitative analysis of the reasoning traces resolves this apparent paradox.
>
> The key explanation lies in the **shift in error severity** as RLVR progresses.
>
> 1.  **Binary Metrics vs. Error Severity:** Our CoT-Pass metric is binary: a single detected error renders the entire trace "0". While effective for measuring capability boundaries, this binary classification masks the *magnitude* of the error.
>     * **Base LLM Errors:** In base models, "incorrect" traces typically contain fatal logic flaws, hallucinations, or complete derailments, as shown in case studies of Appendix A.7. SFT on these traces is harmful.
>     * **Post-RLVR (DAPO) Errors:** In contrast, the "incorrect" traces generated by the DAPO model are fundamentally different. Because the model has already aligned with the correct reasoning distribution, its "failures" are often structurally sound traces containing minor, non-propagating errors.
> 2.  **Benign vs. Fatal Errors:** In practice, we observe that DAPO's "incorrect" traces often contain local errors that do not invalidate the broader reasoning logic.
>     * *Example 1 (Enumeration):* The model might make a minor slip when listing cases but correctly summarizes the properties (e.g., minimum) needed for the next step.
>     * *Example 2 (Derivation):* The model might mis-simplify an intermediate equation but still derive a correct equation for the subsequent logic.
>     * **Why SFT works here:** Because these errors are "benign" (local and non-fatal), the SFT process still extracts high-value signals regarding the *structure* and *depth* of the reasoning process. The model learns the valid "skeleton" of the solution, even if the specific sample contained a "typo-level" logical error.
> 3.  **Verifier Reliability:** This does not imply the verifier is unreliable; rather, it confirms the verifier is strict. The verifier correctly flags these minor slips as errors (high precision). However, the *training value* of a reasoning trace is not binary. A trace with 99% correct logic and one minor slip is a highly effective training sample for Pass@1, even if it strictly fails CoT-Pass verification.
>
> **Conclusion:** The observation that SFT on DAPO's "incorrect" traces improves performance is not evidence of verifier failure, but evidence of **RLVR success**. It demonstrates that RLVR has transformed the model's failure mode from "logical incoherence" to "structurally correct reasoning with minor execution slips," which remains highly robust for learning final answer prediction.

---

> ### Author Response · Authors · 2025-11-20
>
> ### Response to Question 2: SFT Influence and Model Comparison
>
> > It would be good to see the influence of SFT on both Pass@K and CoT-Pass@K. For example, DAPO-Qwen-32B and DeepSeek-R1-Distilled-Qwen-32B have similar avg@32 on AIME24. It would be interesting to see which model performs better when using CoT-Pass@K as a metric.
>
>
> We appreciate this suggestion. However, we respectfully point out a factual discrepancy regarding the performance parity of these models, which makes the proposed comparison scientifically inequitable.
>
> * **Performance Gap:** Contrary to the premise that the models have similar performance, `DeepSeek-R1-Distill-Qwen-32B` significantly outperforms `DAPO-Qwen-32B` on AIME 2024. As reported in its technical report (https://arxiv.org/pdf/2501.12948), the Distilled-32B model achieves an accuracy of **72.6%**, whereas `DAPO-Qwen-32B` (trained via pure RLVR) achieves **50.0%**.
> * **Distillation vs. Pure RL:** This large gap exists because `DeepSeek-R1-Distill-Qwen-32B` is distilled from a **671B parameter teacher model** (DeepSeek-R1), allowing it to inherit vast knowledge and superior logical priors directly. In contrast, `DAPO-Qwen-32B` attempts to self-evolve reasoning purely from a 32B base model via reinforcement learning.
> * **Conclusion:** Since the Distilled model starts with a "teacher-provided" advantage that fundamentally alters its CoT distribution, comparing its `CoT-Pass@K` dynamics to a pure RLVR model would conflate the effects of *distillation* with the effects of *SFT*. Therefore, we believe this specific comparison would not yield a fair or isolated insight into the impact of SFT on self-generated reasoning.

---

> ### Comment · Reviewer_jCat · 2025-11-24
> **Follow up**
>
> I thank the authors' efforts in addressing my concerns. For weakness 1 and question 1, I appreciate the new results on additional verifier (gpt-oss). These results have significantly strengthen the overall soundness of the manuscript. For weakness 2, the interpretation make sense for me. For my previous question 2, apologies, I mixed up the performance of the R1-distilled-32B with other models.
>
> After carefully gone through other reviewers' opinions, the authors' response and the updated manuscripts, my previous major concerns are well addressed. Therefore, I raised my score to 8 to champion this work.
>
> Additionally, I have an open question that I would like to discuss with the authors. Does the significant discrepancy between CoT-Pass@K and Pass@K suggest that the rewarding in popular RLVR algorithms, such as GRPO, is very coarse? In many cases, the final answer is correct even when the reasoning process contains substantial errors. Could CoT-Pass@K provide more insights for designing novel RL algorithms, particularly regarding more effective credit assignment? I believe discussion on this point could maximize the impact of this work.

---

> > ### Author Response · Authors · 2025-11-25
> > **Response to Follow-Up Open Questions**
> >
> > We sincerely thank the reviewer for their thorough engagement with our rebuttal, for examining the other reviews, and for the decision to champion this work. We are delighted that the additional results on the `gpt-oss` verifier and our responses have successfully addressed your major concerns.
> >
> > Regarding your open question on the discrepancy between **CoT-Pass@K** and **Pass@K**, and the potential coarseness of rewards in algorithms like GRPO, we find this question remarkably insightful. It touches upon a fundamental challenge in current RLVR research, and we agree with your assessment that purely using verifiable rewards (outcome supervision) can be coarse. While such rewards provide a strong signal to elicit long CoT behaviors, they do not **explicitly** guarantee the correctness or quality of the reasoning process itself (so in this paper, we call it as implicitly incentivzing correct reasoning).
> >
> > **1. The Coarseness of Outcome Supervision**
> >
> > As you noted, models may arrive at the correct final answer despite remaining errors in the reasoning chain. This aligns with observations of the **R1-Zero** approach (introduced in the DeepSeek-R1 technical report), which utilizes GRPO on base models. While effective at boosting performance, R1-Zero is known to suffer from poor readability and language mixing. Our work extends this finding, revealing that beyond readability issues, these models often hide incorrect mathematical knowledge, flawed derivations, and logical gaps behind correct final answers.
> >
> > This phenomenon is not unique to free-form math. In other verifiable domains—such as competitive coding, theorem proving, or software engineering—verification confirms the end-state but not the efficiency, readability, or logical soundness of the intermediate steps. As OpenAI noted (https://cookbook.openai.com/articles/gpt-oss/handle-raw-cot), raw CoT data contains potentially harmful information, which aligns with why outcome-based RLVR improves performance but does not guarantee high-quality, readable intermediate reasoning.
> >
> > **2. Addressing the Limitation: A Holistic Approach**
> >
> > We agree with you that although RLVR implicitly incentivizes correct reasoning, it has troubles when answer correctness is met while CoT correctness is not achieved yet. To bridge the gap between `Pass@K` and `CoT-Pass@K`, we believe the community must look beyond the pure RL paradigm toward a holistic system:
> >
> > - **Pre-Training and Model Scaling:**
> >
> >   High-quality pre-training is indispensable. In our theoretical framework, a strong base model satisfies the logic prior assumption: $P(\text{Correct Answer} \mid \text{Correct CoT}) - P(\text{Correct Answer} \mid \text{Incorrect CoT}) > 0$. Larger models generally possess better discrimination between correct (perfect) and incorrect (imperfect) CoTs, allowing them to generate higher-quality reasoning implicitly during RLVR. This offers a perspective on why R1-Zero applied to DeepSeek-V3-Base (600B) achieves 71.0% Pass@1 on AIME 2024, whereas Qwen-32B-Base achieves only 47.0%. The 600B model likely has lower perplexity regarding human logic and fewer internal knowledge errors, providing a better starting condition for RLVR.
> >
> > - **Specialized Mid-Training (Cold Start):**
> >
> >     We emphasize the importance of a "cold-start" phase before RLVR, similar to the standard DeepSeek-R1 recipe. Fine-tuning the base model on carefully curated, high-quality reasoning traces helps mitigate knowledge hallucinations and logic errors early on. This stage ensures the model is "on distribution" for high-quality reasoning before the "coarse" exploration of RL begins. This view could also explain why DeepSeek-R1 achieves 79.8% on AIME 2024 while DeepSeek-R1-Zero only obtains 71.0%.
> >
> > - **Novel RLVR Algorithms Beyond Answer Supervision:**
> >
> >     To close the gap further, we see some avenues for algorithmic innovation:
> >     1. **More Challenging Questions:** We can tolerate imperfect CoTs initially but continuously introduce more challenging problems with verifiable answers. This forces the model to "grow" its capabilities to solve harder tasks where "lucky guesses" or flawed logic are less likely to yield correct answers.
> >     2. **Process Supervision & Rubrics:** We can move beyond binary outcome rewards. Designing systematic rubric systems or Process Reward Models (PRMs) that can assign credit to intermediate steps is crucial. `CoT-Pass@K` offers insight into per-step correctness, but future metrics must also evaluate conciseness, penalize over-thinking or repetition, and ensure language consistency without triggering reward hacking.
> >
> > In summary, we believe that a closed-loop system integrating improved pre-training, high-quality mid-training, and improved RLVR is the path toward robust reasoning LLMs.
> >
> > Thank you again for this stimulating discussion. We believe these implications are vital for the future of reasoning research and are grateful for the opportunity to elaborate on them.

---

### Official Review · Reviewer_BkYC · 2025-10-31

**Soundness:** 2
**Presentation:** 3
**Contribution:** 2
**Rating:** 4
**Confidence:** 3

**Summary:**

* **Starting Point**: The paper aims to argue that **RLVR truly enhances the model's capabilities**
* **New Metric**: The paper proposes the **CoT-Pass@K** metric, which is better suited than the commonly used Pass@K to demonstrate the **model's capability improvement after RLVR**
* **Experimental Validation**: The experiments verify that for multiple math tasks, **CoT-Pass@K better highlights the performance gap** in model capabilities compared to Pass@K

**Strengths:**

1. The paper reads smoothly and presents its ideas in a clear, easy-to-follow style.
2. The experimental design is comprehensive and well thought-out.
3. RLVR provides only marginal gains at Pass@1024 or Pass@2048, but demonstrably boosts Pass@1—a widely recognized pattern. This supports the prevailing view that RLVR achieves its effect by learning correct reasoning, a claim for which your experiments offer some evidence.

**Weaknesses:**

Please refer to the "Questions" section for details.

**Questions:**

1. Why opt for a smaller model to judge the reasoning paths? Doesn't the accuracy of the judgment become constrained by the capacity of the referee model? Should a more powerful referee model be employed instead?

2. For tasks where a direct reward signal is unavailable in standard RL training, the common approach is often to use a rubric combined with a powerful evaluation model to assign rewards. Given this, could your CoT-Pass@K metric be integrated into the RLVR training process to yield beneficial results?

3. Based on the experimental results, we observe that the CoT-Pass@K score for 'no-thinking' models is generally suppressed compared to their Pass@K score. However, this method does not appear to provide a significantly better evaluation metric for RLVR models (as the difference between Pass@K and CoT-Pass@K for RLVR seems minor). Therefore, what are the practical applications or use cases for your proposed method?

---

> ### Author Response · Authors · 2025-11-20
> **Thank you for your feedback**
>
> ### Response to Comments on Pass@K Gains (Strength 3)
>
> > RLVR provides only marginal gains at Pass@1024 or Pass@2048, but demonstrably boosts Pass@1—a widely recognized pattern.
>
> We appreciate this observation regarding the saturation of Pass@K at higher sample counts. However, we wish to offer an important clarification regarding the scope and interpretation of this phenomenon.
>
> - **Distinction Between Code and Math:** We respectfully point out that the observation of "marginal gains at high K" is specific to *mathematical reasoning*. In the domain of **code reasoning**, both our experimental results (Section 3.2) and independent third-party studies ([https://arxiv.org/abs/2505.16400](https://arxiv.org/abs/2505.16400)) confirm that RLVR yields steady improvements in Pass@K up to $K=1024$.
> - **The Unreliability of Standard Pass@K in Math:** For math benchmarks, we argue that standard Pass@K is a noisy metric that fails to accurately represent the model's capability boundary. A central contribution of this work is demonstrating that this metric can be inflated by "lucky guesses." By introducing **CoT-Pass@K**, we highlight the necessity of evaluating the correctness of the *reasoning process* alongside the final answer.
> - **Theoretical Mechanism:** Finally, we go beyond simply reporting these metrics to explain the underlying cause. We provide a theoretical framework (supported by our training dynamics analysis) demonstrating that RLVR **implicitly incentivizes correct reasoning paths**. This confirms that the improvements we observe are driven by genuine learning of logical structures rather than mere answer overfitting.
>
> ### Response to Question 1: The Validity of Verification
>
> > Why opt for a smaller model to judge the reasoning paths? Doesn't the accuracy of the judgment become constrained by the capacity of the referee model? Should a more powerful referee model be employed instead?
>
> We appreciate these critical questions. We understand the concern regarding the capacity gap between the 8B verifier and the 32B target model. To address this, we have conducted additional experiments and manual audits to ensure our findings are robust.
>
> - **Validation via Stronger Verifiers:** Following your suggestion, we introduced `gpt-oss-20b` and `gpt-oss-120b` (utilizing high-reasoning effort mode) as external verifiers. As illustrated in **Figure 11 (Appendix A.8)**, the results across these diverse verifiers corroborate our original claims. While `gpt-oss-120b` is naturally stricter in identifying subtle logical errors, the relative trends and the efficacy of RLVR remain consistent. This confirms that the improvements measured by CoT-Pass@K are not artifacts of a capacity-constrained verifier.
> - **Extensive Manual Verification:** To further validate the automated judgments, we performed a rigorous manual inspection of the data. **Appendix A.7** details representative examples from AIME 24-25 where base LLMs arrive at correct answers via incorrect reasoning. We emphasize that these cases are **not cherry-picked**; as shown in the comprehensive summaries in **Tables 2 and 3** (the last page in Appendix), these specific questions correspond to high Pass@K ($K=1024$) but zero CoT-Pass@K ($K=1024$). This manual ground-truth confirms that the 8B verifier is correctly penalizing "lucky guesses" and heuristics, aligning with human knowledge and logic.
> - **Justification for the 8B Verifier:** Finally, we clarify why `DeepSeek-R1-Qwen3-8B` remains an effective verifier despite its smaller parameter count:
>     - **Distillation:** The model is distilled from a 600B teacher specifically for mathematical reasoning, inheriting high-level verification capabilities on math problems.
>     - **Test-Time Scaling:** In reasoning-heavy tasks, effective inference capacity is a function of *parameters $\times$ generated tokens*. By utilizing a long-context reasoning mode (often generating 40k-64k tokens), the 8B model "unrolls" significant computation along the sequence dimension. This test-time scaling allows it to decompose and verify the reasoning of the 32B model (which typically operates with a much shorter context window) effectively.

---

> ### Author Response · Authors · 2025-11-20
>
> ### Response to Question 2: Rubrics as Reward and CoT-Pass@K
>
> > For tasks where a direct reward signal is unavailable in standard RL training, the common approach is often to use a rubric combined with a powerful evaluation model to assign rewards. Given this, could your CoT-Pass@K metric be integrated into the RLVR training process to yield beneficial results?
>
> We appreciate this interesting perspective on integrating rubric-based metrics into the training loop.
>
> - **Scope of RLVR vs. AI Feedback:** We agree that using model-based rubrics as reward signals is essential for tasks where deterministic verification is unavailable (e.g., open-ended writing). However, the specific scope of this work is Reinforcement Learning with **Verifiable** Rewards (RLVR). Our primary research goal was to determine if RLVR can extend the reasoning capability boundary of LLMs using only *objective, ground-truth outcome signals* (answer correctness).
> - **Implicit Optimization of Correct Reasoning:** A key finding of our paper is that we demonstrate—both theoretically and empirically—that optimizing for simple answer correctness **implicitly optimizes for reasoning correctness**. The pre-trained model learns that the most reliable path to a correct answer is a correct reasoning chain. This explains how RLVR works for LLMs to bypass the challenge of providing precise and effective supervision on long-form, complex CoTs.
> - **Role of CoT-Pass@K:** We designed CoT-Pass@K primarily as an **evaluation metric** to address the specific noise in Math benchmarks, where models occasionally arrive at correct answers via incorrect logic (false positives) for hard problems. In contrast, for domains like Code, where execution provides a deterministic check, standard Pass@K remains a reliable measure.
> - **Future Work:** While our theoretical framework currently relies on the existence of a ground-truth verification signal, extending these mechanisms to unverifiable domains—where a rubric or stronger model serves as the proxy for truth—is a valuable direction. We believe our insights into implicit reasoning optimization could inform how effective such rubric-based rewards might be, but we leave this exploration for future work.
>
> ### **Response to Question 3: Practical Utility of CoT-Pass@K**
>
> > Based on the experimental results, we observe that the CoT-Pass@K score for 'no-thinking' models is generally suppressed compared to their Pass@K score. However, this method does not appear to provide a significantly better evaluation metric for RLVR models (as the difference between Pass@K and CoT-Pass@K for RLVR seems minor). Therefore, what are the practical applications or use cases for your proposed method?
>
> We appreciate this opportunity to clarify the intended role and implications of our proposed metric.
>
> - **Metric vs. Method:** First, we wish to distinguish that **CoT-Pass@K** is not a training method or application, but a rigorous **evaluation metric**. While standard `Pass@K` measures only answer correctness, `CoT-Pass@K` requires the correctness of both the final answer *and* the reasoning process. Consequently, the quantitative gap between these two metrics serves as a proxy for "lucky guesses"—instances where the model arrives at the correct answer through flawed reasoning.
> - **Diagnosing Base Models (The "Gap"):** For base ("no-thinking") models, the significant suppression of `CoT-Pass@K` relative to `Pass@K` is a critical finding. It reveals that standard metrics **inflate** the perceived reasoning capability of base models. Our validation with stronger verifiers (`gpt-oss-120b`) confirms that these models frequently "guess" or exploit heuristics for hard questions. Therefore, `CoT-Pass@K` is essential for accurately defining the true reasoning capability boundary, which standard metrics obscure.
> - **Defending RLVR:** We argue that the observation that `Pass@K` and `CoT-Pass@K` converge (i.e., the gap is minor) for RLVR models is **not a limitation of the metric, but the primary validation of our theory**. This convergence empirically demonstrates our core claim: RLVR implicitly incentivizes correct reasoning. It shows that under RLVR, the model learns that the most reliable path to a correct answer (reward) is a correct reasoning chain, effectively eliminating "guessing" behavior.
> - **Scientific Impact:** The primary contribution of this work is not to propose a specific method, but to establish a clearer scientific understanding of how Pre-training and RLVR interact. By using `CoT-Pass@K` to strip away the noise of lucky guesses, we unveil the "real" effect of RLVR on math reasoning: it does not merely boost scores, it fundamentally transforms the model's generation process from unstable matching to rigorous derivation.

---

> ### Author Response · Authors · 2025-11-27
> **A Kind Reminder**
>
> Dear Reviewer BkYC,
>
> Thank you again for your thoughtful review. We have now submitted our response.
>
> Could you please check it at your convenience to see if your concerns have been adequately addressed?
>
> We are happy to provide further clarification if you have any additional questions.

---

### Official Review · Reviewer_RnmQ · 2025-11-01

**Soundness:** 3
**Presentation:** 3
**Contribution:** 3
**Rating:** 6
**Confidence:** 3

**Summary:**

This paper investigates whether Reinforcement Learning with Verifiable Rewards (RLVR) genuinely improves the reasoning capability of large language models, rather than merely increasing sampling efficiency. It provides both theoretical analysis and empirical evaluation to show that RLVR implicitly incentivizes correct reasoning chains. The authors introduce a new evaluation metric, CoT-Pass@K, which verifies both intermediate reasoning correctness and final answers, and demonstrate that GRPO-based RLVR improves reasoning quality across multiple math and code benchmarks. Results suggest that RLVR not only reinforces correctness but also enhances reasoning consistency and generalization.

**Strengths:**

1. The paper provides a well-structured theoretical analysis explaining how GRPO gradients implicitly favor logically consistent reasoning traces, offering insights on why RLVR often improves reasoning without explicit step-wise rewards.

2. Experiments on multiple reasoning domains (math and code) and models (Qwen, Nemotron) show consistent gains under both Pass@K and the proposed CoT-Pass@K metrics. The analyses of training dynamics and reasoning correctness lend strong empirical support to the theoretical claims.

3. The new CoT-Pass@K metric could be a useful tool for the community, bridging the gap between evaluating correctness of answers and correctness of reasoning.

**Weaknesses:**

1. The reliability of CoT-Pass@K hinges on judgments from a single verifier (DeepSeek-R1-Qwen3-8B). While multi-verification is discussed, evaluating under different verifiers would improve confidence in these results. How robust are CoT-Pass@K results when using different verifiers? -- we can fix the verifier used in training and just use different verifiers for evaluation.

2. Potentially questionable assumptions. The theoretical analysis assumes that reasoning traces with correct answers have systematically higher probabilities, which may not always hold in domains like commonsense or multi-hop reasoning. The practical validity of this assumption could be discussed more thoroughly. Do we have any empirical examples?

3. While the paper shows improved correctness, it does not analyze whether RLVR reduces reasoning diversity or leads to overconfident, deterministic reasoning traces. Add some quantitative analysis of reasoning diversity would further strengthen the paper.

**Questions:**

- Could RLVR’s implicit reasoning incentive be combined with explicit step-wise rewards to further boost reasoning reliability?
- It'd be great to see more details on training cost/efficiency and scaling behavior

---

> ### Author Response · Authors · 2025-11-20
> **Thank you for your feedback**
>
> ### Response to Weakness 1: Robustness of Verifiers and CoT-Pass@K
>
> > The reliability of CoT-Pass@K hinges on judgments from a single verifier (DeepSeek-R1-Qwen3-8B). While multi-verification is discussed, evaluating under different verifiers would improve confidence in these results. How robust are CoT-Pass@K results when using different verifiers? -- we can fix the verifier used in training and just use different verifiers for evaluation.
>
> We appreciate this constructive suggestion. To address your concern regarding the reliance on a single verifier, we have expanded our evaluation to include larger, more capable models.
>
> - **Robustness across stronger verifiers:** We followed your advice to evaluate our results using `gpt-oss-20b` and `gpt-oss-120b` (specifically in high-reasoning effort mode). As illustrated in **Figure 11 (Appendix A.8)**, the results using these diverse verifiers are consistent with our original findings. This confirms that the `CoT-Pass@K` trends reported in the paper are robust and not artifacts of a specific verifier model.
> - **Rigorous Manual Verification:** Beyond automated metrics, we validated our findings through extensive manual inspection. **Appendix A.7** details representative examples from AIME 24-25 where base LLMs produce correct answers through incorrect reasoning. Importantly, these cases are **not cherry-picked**; as shown in **Tables 2 and 3** (last page in Appendix), these specific questions correspond to high Pass@K ($K=1024$) but low CoT-Pass@K ($K=1024$). This manual verification confirms that `Qwen-32B-Base` frequently "guesses" or exploits heuristics to reach correct answers, whereas RLVR incentivizes valid reasoning chains that even strict verifiers (like `gpt-oss-120b`) find error-free.
> - **Justification for the 8B Verifier:** Finally, we clarify why `DeepSeek-R1-Qwen3-8B` (DS-8B) serves as an effective verifier despite its size.
>     - **Distillation:** DS-8B is distilled from a 600B teacher model, inheriting significant reasoning capabilities on benchmarks like AIME.
>     - **Test-Time Scaling:** More importantly, the model utilizes extensive test-time computation. With a response length often exceeding 40K tokens (up to 64K), the effective inference capacity is a function of both parameters and generated tokens. By "stacking" the model along the sequence dimension during autoregressive generation, the 8B model achieves a reasoning depth sufficient to verify the 32B base model.

---

> ### Author Response · Authors · 2025-11-20
>
> ### Response to Weakness 2: Validity of the Logical Prior assumption
>
> > Potentially questionable assumptions. The theoretical analysis assumes that reasoning traces with correct answers have systematically higher probabilities, which may not always hold in domains like commonsense or multi-hop reasoning. The practical validity of this assumption could be discussed more thoroughly. Do we have any empirical examples?
>
>
>
> We thank the reviewer for this helpful comment. Before addressing the assumption itself, we would like to briefly clarify our formulation of the *Logical Coherence* assumption. Our assumption does **not** require that correct answers have higher probability than incorrect ones. (i.e.,	$P(Correct Answer) > P(Incorrect Answer)$).
>
> Instead, the assumption is **conditional**:
>
> $P(Correct Answer∣Correct CoT)  >  P(Correct Answer∣Incorrect CoT)$
>
> That is, we assume only that *given* a logically coherent reasoning trace, the model is more likely to reach the correct answer than when its reasoning contains explicit errors.
>
> With this clarification, we agree that this assumption does not hold universally; indeed, we view violations of this assumption as a primary reason why even leading reasoning models continue to make mistakes across math, coding, and commonsense domains.
>
> However, we defend this assumption as a fundamental property of **effective pre-training**:
>
> - **Theoretical Intuition:** We posit that this assumption is a necessary condition established by high-quality pre-training. Intuitively, for a high-capacity model trained on massive, high-quality corpora, the conditional probability of generating the correct answer *must* be higher given a correct CoT than given an incorrect CoT with erroneous intermediate steps. Our theorem dictates that if this condition is not met, the gradient direction will fail to guide the model toward generating improved reasoning traces in subsequent rounds.
> - **Empirical Evidence:** In practice, we observe strong evidence for this assumption. When prompting a base LLM to solve medium-difficulty problems, correct reasoning traces consistently lead to the correct answer. Conversely, when the model follows an incorrect reasoning path, the probability of the token corresponding to the *correct* answer typically drops to near zero. This indicates that the pre-trained model rarely "accidently" samples the correct answer given a flawed derivation.
> - **Nuance in Later Stages:** We also observe that the validity of this assumption evolves during training. As RLVR progresses, the model's reasoning becomes mostly correct, containing only subtle intermediate errors. In these cases, the assumption may weaken because the probability gap between a "perfect" CoT and a "mostly correct" CoT shrinks. This phenomenon aligns with the convergence behavior we observe in practice: when the model can no longer easily distinguish between perfect and slightly flawed reasoning based on answer probability alone, optimization plateaus.
>
> In summary, we view the Logical Prior assumption as a critical property guaranteed by robust pre-training, while RLVR serves the indispensable role of leveraging this prior to elicit deeper, more logically consistent reasoning.

---

> ### Author Response · Authors · 2025-11-20
>
> ### Response to Weakness 3: Reasoning Diversity
>
> > While the paper shows improved correctness, it does not analyze whether RLVR reduces reasoning diversity or leads to overconfident, deterministic reasoning traces. Add some quantitative analysis of reasoning diversity would further strengthen the paper.
>
> We appreciate your insightful comments regarding the potential impact of RLVR on reasoning diversity. We agree that the risk of mode collapse—where RLVR leads to overconfident or deterministic traces—is a significant open question in the field. However, we wish to clarify our perspective on this issue relative to the scope of our paper.
>
> - **Primary Focus on Correct Reasoning and Capability Boundaries:** The central motivation of this work is to uncover the fundamental effect of RLVR: improving not just sampling efficiency (Pass@1) but the **reasoning capability boundary** (Pass@K for code, CoT-Pass@K for math). The underlying mechanism is implicitly incentivizing correct reasoning. We demonstrate that this paradigm does not inherently limit the base LLM; rather, with proper training recipes, it extends the frontier of solvable problems. While diversity is an important variable, our focus remains on the correctness and depth of reasoning.
> - **Challenges in Diversity Measurement:** Quantifying "reasoning diversity" remains a non-trivial challenge. Standard statistical measures (e.g., n-gram overlap, semantic embeddings) fail to capture the structural nuances of logical reasoning. Following a review of the literature, we found no consensus on a reliable metric that could definitively characterize reasoning diversity in this context.
> - **Pass@K as a Proxy for Diversity:** Prior studies often utilize a decrease in Pass@K as a proxy for reduced diversity (under the assumption that lower diversity reduces the coverage of solvable problems). We offer a nuanced counter-perspective:
>     - **In Code Reasoning:** While poorly tuned RLVR models often show a drop in Pass@K (indicating diversity loss), we reproduce and confirm that *properly trained* models (as detailed in our work) actually **extend Pass@K** on benchmarks like LiveCodeBench. This suggests that "effective diversity"—the coverage of solvable problems—is maintained.
>     - **In Math Reasoning:** We caution against using standard Pass@K as a diversity proxy here, as it fails to distinguish between correct derivation and lucky guessing. Instead, utilizing our proposed **CoT-Pass@K**, we observe steady improvements in the coverage of correctly *reasoned* solutions on AIME. This indicates that while the model may converge toward specific patterns, it is converging toward *valid* reasoning paths rather than collapsing into degenerate ones.
>
> ### Response to Question 1: Combining Implicit and Explicit Rewards
>
> > Could RLVR’s implicit reasoning incentive be combined with explicit step-wise rewards to further boost reasoning reliability?
>
> We appreciate this insightful question regarding the synergy between outcome and process supervision. We agree that combining the implicit reasoning incentives of RLVR with explicit step-wise rewards (process supervision) is a promising avenue to further enhance reasoning reliability.
>
> However, we emphasize that designing precise, scalable step-wise rewards for complex reasoning tasks remains a substantial challenge. If intermediate reward signals are noisy, sparse, or susceptible to "gaming" (reward hacking), the resulting hybrid model risks underperforming compared to a pure implicit mechanism. Consequently, while we recognize the potential of this approach, we view the development of robust, hack-resistant process rewards as a distinct and valuable direction for future research.
>
> ### Response to Question 2: Cost & Scaling Behavior
>
> > It'd be great to see more details on training cost/efficiency and scaling behavior
>
> We appreciate your interest in the computational demands and scaling properties of our approach.
>
> - **Training Cost Details:** We have documented the specific training costs for our reproduction of `DAPO-Qwen-32B` in **Appendix A.5**. To provide a concrete benchmark: a single complete training run required over two weeks utilizing a cluster of **32 AMD MI300X GPUs**. We have successfully replicated this process multiple times to ensure the stability and reproducibility of our results.
> - **Scaling Behavior:** While we recognize the value of scaling laws, conducting a comprehensive study across orders of magnitude in model size or compute budget is unfortunately beyond the scope of our current academic computational resources.
> - **Scope & References:** As the primary contribution of this work is to investigate the *mechanisms* by which RLVR improves LLM reasoning, we focus our analysis there. For readers interested in the specific scaling laws of RLVR with massive compute resources, we recommend recent studies dedicated to that topic, such as [https://arxiv.org/abs/2510.13786](https://arxiv.org/abs/2510.13786).

---

> ### Author Response · Authors · 2025-11-27
> **A Kind Reminder**
>
> Dear Reviewer RnmQ,
>
> Thank you again for your thoughtful review. We have now submitted our response.
>
> Could you please check it at your convenience to see if your concerns have been adequately addressed?
>
> We are happy to provide further clarification if you have any additional questions.

---

### Author Response · Authors · 2025-11-20
**General Response to All Reviewers**

We thank the reviewers for the considerable time and effort devoted to evaluating our work. We appreciate the insightful questions and constructive suggestions. Below, we summarize the core contributions of this paper and address common concerns raised across reviews.

### 1. Summary of Unique Contributions

First, we highlight the **unique contributions of this work** in the context of the current literature:

- **Demonstrating that RLVR extends the reasoning capability boundary of base LLMs:**
    - At present, most peole still believe RLVR only improves sampling efficiency but does not extend the reasoning capability boundary (measured in Pass@K, [https://neurips.cc/virtual/2025/loc/san-diego/poster/119944](https://neurips.cc/virtual/2025/loc/san-diego/poster/119944)). A few contemporary studies have demonstrated this is not true in code reasoning ([https://arxiv.org/abs/2505.16400](https://arxiv.org/abs/2505.16400)), we reproduce and confirm this in code reasoning, and more fundamentally, extend these findings to **mathematical reasoning** with only answer correctness as the reward.
    - We identify that *reasoning correctness* is a critical confounding factor that has previously obscured the extended capability boundary in math benchmarks. To address this, we propose **CoT-Pass@K**, a metric that jointly evaluates the correctness of both the final answer and the reasoning process.
- **Uncovering a fundamental mechanism: RLVR implicitly optimizes correct reasoning:**
    - We establish a foundational theorem: the gradient of GRPO provably increases the probability of generating correct Chain-of-Thought (CoT) paths from training questions.
    - We analyze the training dynamics and generalization behaviors of the complete RLVR process. Our empirical results align closely with our theoretical derivation.
    - We also discuss the limitations of current RLVR methodologies through this analytical lens.
- **Disentangling the roles of Pre-training and RLVR in reasoning:**
    - **Pre-training** establishes the necessary knowledge base and logical priors (Assumption Eq (4) in Section 4).
    - **RLVR** acts as an indispensable mechanism for *eliciting* correct reasoning (Theorem 1 in Section 4).
    - Crucially, RLVR does not merely select existing correct paths; it incentivizes deeper, longer, and logically superior reasoning compared to the base model.

### 2. Reliability of the Verifier (DeepSeek-R1-0528-Qwen3-8B)

We address the common concern regarding the reliability of using `DeepSeek-R1-0528-Qwen3-8B` (DS-8B) to verify the CoT correctness of larger models like `Qwen-32B-Base` and `DAPO-Qwen-32B`.

- **Validation via Stronger Verifiers:** Following reviewers' advice, we validated our results using larger, more capable models as verifiers, specifically `gpt-oss-20b` and `gpt-oss-120b` (high-reasoning effort mode). As shown in **Figure 11 (Appendix A.8)**, experiments with these diverse verifiers corroborate our main claims. While `gpt-oss-120b` is strictly more rigorous in identifying reasoning errors, the overall trends remain consistent.
- **Extensive Manual Verification (No Cherry-Picking):** We emphasize that our conclusions are supported by rigorous manual verification, particularly for "High Pass@K / Low CoT-Pass@K" cases.
    - **Appendix A.7** details typical examples from AIME 24-25 where base LLMs could produce correct answers via incorrect reasoning.
    - **Tables 2 and 3** provide a comprehensive summary of Pass@K and CoT-Pass@K ($K=1024$) for *every* question in AIME 24-25. Questions with detailed case studies in Appendix A.7 are marked with a *, demonstrating that our qualitative analysis is representative of the dataset and not cherry-picked.
    - We will publicly release all the reasoning traces and verfication results on AIME 24-25 so that anyone can perform per-question and per-response analysis as we did in Appendix A.7 and Tables 2 and 3.
- **Test-Time Scaling and Effective Capacity:** Finally, we clarify why the 8B parameter model (DS-8B) is a sufficient verifier for the 32B base model. The key insight is **test-time scaling**.
    - In reasoning-heavy tasks, effective model capacity during inference is a function of *parameters $\times$ generated tokens because autogressive generation is inherently stacking a next-token prediction model multiple times along the sequence dimension.*
    - DS-8B utilizes a long-response reasoning mode (approx. $8B \times 64K$ tokens), whereas `Qwen-32B-Base` is typically bounded by standard generation lengths ($32B \times 4K$ tokens).
    - Consequently, the 8B model effectively captures reasoning errors in the 32B base model. While verifying `DAPO-32B` (capacity $\leq 32B \times 16K$) is more challenging, we find DS-8B remains relatively effective in practice because `R1-Zero` training on a 32B base inherits significant inherent reasoning errors that the specialized 8B model (distilled from a 600B teacher model) can easily detect.

---

### Meta-Review · Area_Chair_CvKA · 2026-01-08

**Summary:**

The reviewers were generally positive about the paper’s motivation, presentation and empirical evaluation, while several concerns were raised in the review:
* Verifier reliability (RnmQ, BkYC, jCat): All three reviewers expressed concerns regarding the reliability of using a single 8B verifier (DeepSeek-R1-Qwen3-8B) for assessing CoT correctness. Stronger verifiers or multi-verifier validation were suggested to increase confidence in CoT-Pass@K results.
* Scalability and computational cost (RnmQ, BkYC): The reviewers requested more details on the training cost, efficiency, and scaling behavior of RLVR methods, particularly for larger models or high-K evaluation settings.
* Reasoning diversity and overconfidence (RnmQ): The reviewer raised concerns that RLVR could reduce reasoning diversity or lead to overconfident, deterministic reasoning traces. They suggested quantitative analyses to ensure that correctness improvements do not come at the cost of exploration or variability in reasoning.
* Practical utility of CoT-Pass@K (BkYC): The reviewer questioned the practical applications of the CoT-Pass@K metric, noting that while it highlights differences between Pass@K and reasoning correctness, it was unclear whether it provides significantly better evaluation for RLVR models in practice.
* Interpretation of post-SFT Experiments (jCat): The reviewer found the results from fine-tuning on “incorrect” CoTs counterintuitive and difficult to follow. Specifically, they questioned whether minor local errors versus global reasoning failures were properly distinguished, and how this affects Pass@K and CoT-Pass@K metrics.

The original evaluation was mixed: two out of the three reviewers (RnmQ & jCat) suggested weak acceptance (6), while one reviewer (BkYC) gave a weak rejection (4). Upon reviewing the authors’ rebuttal, most of these concerns have been addressed, and therefore we suggest acceptance as a poster.

**Reviewer Concerns:**

Addressed Concerns:
* Verifier reliability: The authors added new evaluation with stronger verifiers (i.e., gpt-oss-20b, gpt-oss-120b) and performed extensive manual audits. These experiments confirmed that CoT-Pass@K trends are robust and not artifacts of a single verifier.
* Interpretation of post-SFT Experiments: The authors clarified that “incorrect” traces in post-RLVR fine-tuning often contain minor, non-fatal errors. These still provide useful learning signals, explaining why fine-tuning on such traces can improve Pass@1 without violating CoT correctness assumptions.
* Practical utility of CoT-Pass@K: The authors clarified that CoT-Pass@K is intended as an evaluation metric to highlight reasoning correctness versus lucky guessing, rather than as a training method. Convergence between Pass@K and CoT-Pass@K under RLVR validates the metric’s purpose.

Outstanding / Partially Addressed Concerns:
* Reasoning diversity and overconfidence: While the authors explained that their focus is on correctness and capability boundaries, quantitative analysis of reasoning diversity remains limited.
* Scalability and computational cost: The authors provided some training cost details and discussed scaling limitations, but a systematic scaling experiments for larger models is missing.

**Reviewer Scores:**

* Reviewer RnmQ: Originally gave a weak acceptance (6). The main concerns regarding verifier reliability, theoretical assumptions, and training cost were largely addressed in the rebuttal. With the additional experiments and clarifications, the reviewer would likely remain positive (6).
* Reviewer BkYC: Initially gave a weak rejection (4). While some reservations about broader applicability and practical impact may remain, the rebuttal addressed key concerns regarding verifier capacity and clarified the role of CoT-Pass@K as an evaluation metric. Therefore, the reviewer might revise their score upward to a weak acceptance (6).
* Reviewer jCat: Initially gave a weak acceptance (6), but explicitly stated their willingness to champion the paper and updated their score to a strong acceptance (8) after engaging with the authors’ rebuttal.

---

### Decision · Program_Chairs · 2026-01-26

Accept (Poster)